



**Two superimposed cold and fresh anomalies enhanced Irminger Sea**
**deep convection in 2016 – 2018**
Patricia ZUNINO[1], Herlé MERCIER[2] and Virginie THIERRY[3].
1 Altran Technologies, Technopôle Brest Iroise, Site du Vernis , 300 rue Pierre Rivoalon, 29200 Brest,
France
2 CNRS, University of Brest, IRD, Ifremer, Laboratoire d'Océanographie Physique et Spatiale (LOPS),
IUEM, ZI de la pointe du diable, CS 10070 - 29280 Plouzané, France
3 Ifremer, University of Brest, CNRS, IRD, Laboratoire d'Océanographie Physique et Spatiale (LOPS),
IUEM, ZI de la pointe du diable, CS 10070 - 29280 Plouzané, France
Corresponding author: patricia.zuninorodriguez@altran.fr



## ABSTRACT

While Earth system models project a reduction, or even a shut-down, of deep convection in the North Atlantic Ocean in response to anthropogenic forcing, deep convection returned to the Irminger Sea in 2008 and occurred several times since then to reach exceptional depths > 1,500 m in 2015 and 2016. In this context, we used Argo data to show that deep convection persisted in the Irminger Sea during two additional years in 2017 and 2018 with maximum convection depth > 1,300 m. In this article, we investigate the respective roles of air-sea flux and preconditioning of the water column to explain this exceptional 4-year persistence of deep convection; we quantified them in terms of buoyancy and analyzed both the heat and freshwater components. Contrary to the very negative air-sea buoyancy flux that was observed during winter 2015, the buoyancy fluxes over the Irminger Sea during winters 2016, 2017 and 2018 were close to climatological average. We estimated the preconditioning of the water column as the buoyancy that needs to be removed (B) from the end of summer water column to homogenize the water column down to a given depth. B was lower for winters 2016 − 2018 than for the mean 2008 − 2015, including a vanishing stratification from 600 m down to ~1,300 m. It means that less air-sea buoyancy loss was necessary to reach a given convection depth than in the mean and once convection reached 600 m little additional buoyancy loss was needed to homogenize the water column down to 1,300 m. We showed that the decrease in B was due to the combined effects of a cooling of the intermediate water (200 − 800 m) and a decrease in salinity in the 1,200 − 1,400 m layer. This favorable preconditioning permitted the very deep convection observed in 2016 − 2018 despite the atmospheric forcing was close to the climatological average.



## 1. INTRODUCTION


The physical and biogeochemical properties of the oceans are experiencing unprecedented changes
as a result of the human activities (IPCC, 2013). For this reason, a challenge for the oceanographic
community is to disentangle the natural and anthropogenic components of the ocean variability.
Under Climate Change scenarios, Earth system models predict a warmer climate, an increase in
freshwater flux into the ocean due to ice melting (Bamber et al., 2018)), a slow-down of the
Meridional Overturning Circulation (Rahmstorf et al., 2015), and a reduction or shut-down of deep
convection in the North Atlantic (Brodeau & Koenigk, 2016). In this paper, we study the recent
evolution of deep convection in the northern North Atlantic.
Deep convection is the result of a process by which surface waters loose buoyancy due to
atmospheric forcing and sink into the interior ocean. It occurs only where specific conditions are met
including large air-sea buoyancy loss and favorable preconditioning (i.e. low stratification of the
water column) (Marshall & Schott, 1999). In the Subpolar North Atlantic (SPNA), deep convection
takes place in the Labrador Sea and in the Irminger Sea (Kieke & Yashayaev, 2015; Pickart et al. 2003)
connecting the upper and lower limbs of the Meridional Overturning Circulation (MOC) and
transferring climate change signals from the surface to the ocean interior.
The observation and description of deep convection is difficult because deep convection happens at
very short space and time scales (Marshall & Schott, 1999) and during periods of severe weather
conditions. The onset of the Argo program (http://www.argo.net/) at the beginning of the 2000s, has
considerably increased the availability of oceanographic data. Although the sampling characteristics
of Argo are not adequate to observe the smaller scales associated with the process itself, this dataset
allowed the description of the overall intensity of the event, and the properties of the water masses
formed in the winter mixed layer (e.g., Yashayaev and Loder, 2017). The challenge now is to evaluate
how deep convection could evolve under climate change.
In the Labrador Sea, deep convection occurs almost every year, yet with different intensity (e.g.,
Yashayaev and Clarke, 2008; Kieke and Yashayaev, 2015). In the Irminger Sea, Argo and moorings
data showed that deep convection happened in the Irminger Sea during winters 2008, 2009, 2012,
2015 and 2016 (Väge et al., 2009; de Jong et al., 2012; Piron et al. 2015; de Jong & de Steur, 2016;
Piron et al. 2017 ; de Jong et al., 2018). Excluding the winter 2009 when the event was possible
thanks to a favorable preconditioning set the winter before (de Jong et al. 2012), all events coincided
with strong atmospheric-forcing (air-sea heat loss). Prior to 2008, only few deep convection events
were reported because the mechanisms leading to it were not favorable (Centurioni and Gould,


2004) and because the observing system was not adequate (Bacon, 1997; Pickart et al., 2003).
Nevertheless, the hydrographic properties from the 1990s suggested that deep convection reached
as deep as 1,500 m in the Irminger Sea during winters 1994 and 1995 (Pickart et al., 2003).
The convection of winter 2015 was the deepest observed in the Irminger Sea since the beginning of
the 21$^{st}$ century (de Jong et al., 2016; Piron et al., 2016). In this work, we show that deep convection
also happened each winter in the Irminger Sea during the period 2016 – 2018. We investigate the
respective role of atmospheric forcing and preconditioning in setting the convection intensity. We
evaluate them in terms of air-sea buoyancy flux and buoyancy content and, for the first time to our
knowledge in this region, we disentangle the relative contribution of salinity and temperature
anomalies to the preconditioning. The paper is organized as follow. The data are described in Sect. 2.
The methodology is explained in Sect. 3. We expose our results in Sect. 4 and discuss them in Sect. 5.
Conclusions are listed in Sect. 6.

## 2. DATA

We used temperature (T), salinity (S) and pressure (P) data measured by Argo floats north of 55°N in
the Atlantic Ocean. These data were collected by the International Argo program
(http://www.argo.ucsd.edu/, http://www.jcommops.org/) and downloaded from the Coriolis Data
Center (http://www.coriolis.eu.org/). Only data flagged as good (quality Control < 3, Argo Data
Management Team, 2017) were considered in our analysis. Potential temperature (θ), density (ρ)
and potential density anomaly referenced to the surface and 1000 dbar ($\sigma_0$ and $\sigma_1$, respectively) were
estimated from T, S and P data using TEOS-10 (http://www.teos-10.org/). As in Zunino et al. (2017),
we define freshwater as: $FW = \frac{35-S}{35}$ .
We used two different gridded products of ocean T and S: EN4 and ISAS. ISAS (Gaillard et al., 2016;
Kolodziejczyk et al., 2017) is produced by optimal interpolation of *in situ* data. It provides monthly
fields, at 152 depth levels, at 0.5° resolution, from 2002 to 2015. Near real time data are also
availaible for 2016 and 2018. EN4 (Good et al., 2013) is an optimal interpolation of *in situ* data; it
provides monthly T and S at 1° spatial resolution and at 42 depth levels, for the period 1900 to
present.
Net air-sea heat flux (Q), evaporation (E), precipitation (P), wind stress ( $\tau_x$ and $\tau_y$) and sea surface
temperature (SST) data were obtained from ERA-Interim reanalysis (Dee et al., 2011). ERA-Interim
provides data with a time resolution of 12h and a spatial resolution of 0.75°, respectively. The air-sea
freshwater flux (FWF) was estimated as E - P.



We used monthly Absolute Dynamic Topographic (ADT), which was computed from the daily 0.25° -
resolution ADT data provided by CMEMS (Copernicus Marine and Environment Monitoring Service,
http://www.marine.copernicus.eu).

**3. METHODS**
3.1 Quantification of the deep convection
In order to characterize the convection in the Irminger Sea and Labrador Sea in winters 2015-2018,
we estimated the mixed layer depths (MLD) for all Argo profiles collected in the SPNA north of 55°N
from 1$^{st}$ January to 30$^{th}$ April of each year (Fig. 1). Following Piron et al. (2016), the MLD was
estimated by the threshold method (de Boyer Montégut et al., 2004) and the split and merge
method (Thomson and Fine, 2003) complemented by visual inspection of the vertical profiles of ρ
when the two estimates differed. In this paper, deep convection is characterized by profiles with
MLD deeper than 700m (colored big points in Fig. 1) because it is the minimum depth that should be
reached for Labrador Sea Water (LSW) renewal (Yashayaev et al., 2007; Piron et al. 2016).
The winter MLD and the associated θ, S and ρ properties were examined for the Labrador Sea and
the Irminger Sea, separately. We used 48°W as the limit between the Irminger Sea and the Labrador
Sea. We computed the maximum MLD and the MLD third quartile (Q$_3$) (we only used MLD greater
than 700m for the computation of Q$_3$). Q$_3$ is the MLD value that is exceeded by 25% of the profiles
and is equivalent to the aggregate maximum depth of convection defined by Yashayaev and Loder
(2016). The properties (ρ, θ and S) of the mixed layers formed each winter were defined as the
vertical mean from 200 m to the MLD of the n profiles with MLD deeper than 700 m.
3.2. Time series of atmospheric forcing
The air-sea buoyancy flux (B$_{surf}$) was calculated as the sum of the contributions of Q and FWF (Gill,
1982; Billheimer & Talley, 2013). It reads:
$B_{surf} = \frac{\alpha.g}{\rho_0.c_p}.Q - \beta.g.SSS.FWF$         Eq. (1)
Where α and β are the coefficients of thermal and saline expansions, respectively, estimated from
surface T and S.  The gravitational acceleration g is equal to  9.8 m s$^{-2}$, the reference density of sea
water ρ$_0$ is equal to 1026 kg m$^{-3}$ and  heat capacity of sea water C$_p$ is equal to 3990 J kg $^{-1}$ °C$^{-1}$. SSS is
the sea surface salinity. Q and FWF are in W m$^{-2}$ and m s$^{-1}$, respectively.



For easy comparison with previous results that only considered the heat component of the total air-
sea flux (e.g. Yashayaev & Loder, 2017; Piron et al. 2017; Rhein et al., 2017), $B_{surf}$, in $m^2\ s^{-3}$, was
converted to $W\ m^{-2}$ following Eq. (2) and noted $B_{surf}^{*}$
$B_{surf}^{*} = \frac{\rho_0 \cdot c_p}{g \cdot \alpha} \cdot B_{surf}$           Eq. (2)
The FWF was also converted to $W\ m^{-2}$ using:
$FWF^* = FWF \cdot \beta \cdot SSS \cdot \frac{\rho_0 \cdot c_p}{\alpha}$        Eq. (3)
We also computed the horizontal Ekman buoyancy flux ($BF_{ek}$), and their components the horizontal
Ekman heat flux ($HF_{ek}$) and the horizontal Ekman salt flux ($SF_{ek}$), for which $BF_{ek} = SF_{ek} - HF_{ek}$.
$BF_{ek} = -g \cdot (U_e \partial_x SSD + V_e \partial_y SSD) \cdot \frac{c_p}{\alpha \cdot g}$     Eq. (4)
$HF_{ek} = -(U_e \partial_x SST + V_e \partial_y SST) \cdot \rho_0 \cdot C_p$     Eq. (5)
$SF_{ek} = -(U_e \partial_x SSS + V_e \partial_y SSS) \cdot \frac{\beta \cdot \rho_0 \cdot C_p}{\alpha \cdot}$     Eq. (6)
where $U_e$ and $V_e$ are the eastward and westward components of the Ekman horizontal transport
estimated from the wind stress meridional and zonal components. SSD, SST and SSS are ρ, T and S at
the surface of the ocean. The factors on the right of equations are used to obtain $BF_{ek}$, $HF_{ek}$ and $SF_e$ in
$J\ s^{-1}\ m^{-2}$. Because ERA-Interim does not supply SSD or SSS, the monthly S at 5 m depth from EN4 were
interpolated on the same time and space grid as the air-sea fluxes from ERA-Interim (12h and 0.75°,
respectively). SSD was estimated from SST of ERA-Interim and interpolated S of EN4. Data at points
shallower than 1000 m were excluded from the analysis to avoid regions covered by sea-ice.
Following Piron et al. (2016), the time series of atmospheric forcing were estimated for the Irminger
Sea and the Labrador Sea as follows. First, the gridded air-sea flux data and the horizontal Ekman
fluxes were averaged over the region, pink and cyan boxes in Fig. 1 for the Irminger Sea and Labrador
Sea, respectively. The boxes, which are representative of the convection regions, were defined to
include most of Argo profiles with MLD deeper than 700 m and the minimum of the monthly ADT for
either the Irminger Sea or the Labrador Sea. Second, we estimated the accumulated fluxes from 1
September to 31 August the year after. Finally, we computed the time series of the anomalies of the
accumulated fluxes from 1 September to 31 August with respect to the 1993 – 2016 mean.
Finally, in order to quantify the intensity of the atmospheric forcing over the winter, we computed
estimates of air-sea fluxes accumulated from 1 September to 31 March the year after. The associated





errors were calculated by Monte Carlo simulation as the standard deviation of 50 estimates of Q,
FWF and $B_{surf}$ perturbed by one standard deviation of their spatial mean. The error amounted to 0.05
J m$^2$ , 0.04 and 0.03 J m$^{-2}$ for $B_{surf}$*, Q and FWF*, respectively. The error of the horizontal Ekman
transport was also estimated by Monte Carlo simulation and amounted to 0.04 J m$^{-2}$.
## 3.3. Preconditioning of the water column
The preconditioning of the water column was evaluated in terms of buoyancy that has to be
removed $(B(zi))$ to the late summer density profile to homogenize it down to a depth *zi*. It reads:
$$B(zi) = \frac{g}{\rho_0} * \ \sigma_0(zi) * zi \ - \ \frac{g}{\rho_0} \int_{z_i}^{o} \sigma_0 \ (z) dz \qquad \text{Eq. (7)}$$
$\sigma_0$(z) is the vertical profile of potential density anomaly estimated from the profiles of T and S
measured by Argo floats in September in the given region (pink or cyan box in Fig. 1).
Following Schmidt and Send (2007), we split B into a temperature $(B_\theta)$ and salinity $(B_S)$ term as:
$$B_\theta(zi) = -( \ g * \alpha * \ \theta(zi) * zi - \ g * \alpha * \int_{zi}^{o} \theta(z) \ dz) \qquad \text{Eq. (8)}$$
$$B_S(zi) = g * \beta * \ S(zi) * zi \ - \ g * \beta * \int_{zi}^{o} S \ (z) dz \qquad \text{Eq. (9)}$$
In order to compare the B with the heat to be removed and/or air-sea heat fluxes, the buoyancy
results in m$^2$ s$^{-2}$ were converted to J m$^{-2}$.
B, $B_\theta$ and $B_S$ were estimated for a given year from the mean of all September profiles of B, $B_\theta$ and
$B_S$ . The associated errors were estimated as std(B)/√n, where n is the number of profiles used to
compute the September mean values.

**4. RESULTS**
## 4.1. Intensity of deep convection and properties of newly formed LSW
We examine the time-evolution of the Irminger Sea winter mixed layer since the exceptional
convection event of winter 2015 (W2015 hereinafter) (Table 1 and Fig. 3). In W2015 we recorded a
maximum MLD of 1,715 m south of Cape Farewell (Fig. 1a), in line with Piron et al. (2017). The
maximum MLD of 1,575 m observed for W2016 (Fig. 1b) is compatible with the MLD > 1,500 m
observed in a mooring array in the central Irminger Sea by de Jong et al. (2018). We additionally
showed that for both winters Q3 was about 1300 m (Table 1). Now, we describe the convection of



W2017 and W2018. In W2017, deep convection was defined from four Argo profiles in the Irminger
Sea (see Fig. 1c and Fig. 2a-c). The maximum MLD of 1,400 m was observed on 16$^{th}$ March 2017 at
56.65°N − 42.30°W. The aggregate maximum depth of convection Q3 coincided with the maximum
MLD because the estimates are based on only four profiles. In W2018, ten profiles showed MLD
deeper than 700 m in the Irminger Sea (Fig. 1d, 2d-f). The maximum MLD of 1,300 m was observed
on 24 February at 58.12°N, 41.84°W. The aggregate maximum depth of convection Q3 was 1,100 m.
Float 5903102, which was localized South of Cape Farewell, did not profile deeper than 1,100 m in
any of its six cycles (see Fig. 2d-f); it is therefore possible that the MLD was deeper than 1,100 m in
these profiles. Excluding the data of Float 5903102, the aggregate maximum depth of convection Q3
is 1,300 m. These results reveal that convection deeper than 1,300 m occurred during four
consecutive winters in the Irminger Sea.
The properties ($\sigma_0$, S and $\theta$) of the end of winter mixed layer were estimated for the four winters
(Table 1 and Fig. 3). We observed that, between W2015 and W2018, the water mass formed by deep
convection significantly densified and cooled by 0.019 kg m$^{-3}$ and 0.306°C, respectively (see Table 1).
In the Labrador Sea, Q3 increased from 2015 to 2018 (see Table 1). Deep convection observed in the
Labrador Sea in W2018 was the most intense since the beginning of the Argo era (see Fig. 2c in
Yashayaev & Loder, 2016). From W2015 to W2018, newly formed LSW cooled, salted and densified
by 0.134°C, 0.013 and 0.023 kg m$^{-3}$, respectively (Table 1).
The water mass formed in the Irminger Sea is warmer and saltier than that formed in the Labrador
Sea (Fig. 3); the exception is in W2018 when the characteristics of the water masses formed in each
of the basins are very similar. The deep convection in the Irminger Sea is always shallower than in the
Labrador Sea. Both results are discussed later in Sect. 5.
## 4.2. Analysis of the atmospheric forcing in the Irminger Sea
The seasonal cycles of $B_{surf}$* and Q are in phase and of the same order of magnitude, while the FWF*
is positive and one order of magnitude lower than Q and does not present a seasonal cycle (Fig. S1).
The means (1993 − 2018) of the cumulative sums from 1 September to 31 March of Q, FWF* and
$B_{surf}$* estimated over the Irminger box (Fig. 1) are - 2.52 ± 0.43 x 10$^9$ J m$^{-2}$, 0.31 ± 0.11 x 10$^9$ J m$^{-2}$ and
- 2.26 ± 0.51 x 10$^9$ J m$^{-2}$, respectively. Despite $B_{surf}$* is mainly explained by Q, the accumulated FWF*
amounts to ~10 % of the accumulated Q with opposite sign. The atmospheric forcing estimated in
terms of buoyancy is therefore 10% lower on average than when estimated in terms of heat.



Piron et al. (2016) found that "the wind stress led to an Ekman-induced heat loss that reinforced by
about 10% the heat loss induced by the net air-sea heat fluxes". Here, we considered the buoyancy
flux induced by the Ekman response to the wind stress and we estimated the buoyancy, heat and salt
Ekman fluxes ($BF_{ek}$, $HF_{ek}$ and $SF_{ek}$). The means (1993 - 2018) of the accumulated $BF_{ek}$, $HF_{ek}$ and $SF_{ek}$
from 1 September to 31 March amount to $- 0.0004 \pm 0.04 \times 10^9$ J m$^{-2}$, $0.0446 \pm 0.04 \times 10^9$ J m$^{-2}$, and
$0.0626 \pm 0.04 \times 10^9$ J m$^{-2}$, respectively. So, on average, $HF_{ek}$ and $SF_{ek}$ compensate each other resulting
in an almost zero J m$^{-2}$ $BF_{ek}$. However, for particular years with strong wind stress as it was the case in
2015, there is no such compensation and $BF_{ek}$ is different from 0 (see Fig. 4).
We now compare the accumulated $B_{surf}$* from 1 September to 31 March the year after for the last
four deep convection years. It amounted to $- 3.21 \times 10^9 \pm 0.05$ J m$^{-2}$, $- 2.29 \pm 0.04 \times 10^9$ J m$^{-2}$, $- 2.23 \pm$
$0.05 \times 10^9$ J m$^{-2}$ and $- 2.58 \pm 0.05 \times 10^9$ J m$^{-2}$ for W2015, W2016, W2017 and W2018, respectively. The
cumulative sum of $BF_{ek}$ from 1 September 2014 to 31 March 2015 was $- 0.27 \pm 0.04 \times 10^9$ J m$^{-2}$; the
estimates for the following winters were near 0 J m$^{-2}$. When the $BF_{ek}$ is added to the $B_{surf}$*, the
resulting atmospheric forcing is $- 3.48 \times 10^9 \pm 0.05$ J m$^{-2}$, $- 2.19 \pm 0.04 \times 10^9$ J m$^{-2}$, $- 2.20 \pm 0.05 \times 10^9$ J
m$^{-2}$ and $- 2.57 \pm 0.05 \times 10^9$ J m$^{-2}$ for W2015, W2016, W2017 and W2018, respectively. The estimate
for W2015 is ~30% larger than the estimates for 2016 – 2018. Time series of atmospheric forcing
anomalies in Fig. 4 show that this strongly negative W2015 anomaly of accumulated $B_{surf}$* was
caused by very negative Q and FWF anomalies and a negative $BF_{ek}$ as well. During W2016, W2017
and W2018 however, all atmospheric forcing terms were close to zero.
From these results we conclude that, contrary to the very negative anomaly in atmospheric fluxes
over the Irminger Sea observed for W2015, the atmospheric fluxes were close to the mean during
W2016, W2017 and W2018.
## 4.3. Analysis of the preconditioning of the water column in the Irminger Sea
Our hypothesis is that the exceptional deep convection that happened in W2015 in the Irminger Sea
favorably preconditioned the water column for deep convection the following winters. The time-
evolution of θ, S, $\sigma_1$ and of $\Delta\sigma_1$=0.01 kg m$^{-3}$ layer thicknesses (Fig. 5) show a marked change in the
hydrological properties of the Irminger Sea at the beginning of 2015 caused by the exceptional deep
convection that occurred during W2015 (see also Piron et al., 2017). The intermediate waters (500 –
1,000 m) became colder than the years before and, despite a slight decrease in salinity, the cooling
caused the density to increase (Fig. 5c). Fig. 5d shows $\Delta\sigma_1$=0.01 kg m$^{-3}$ layer thicknesses larger than
600 m appearing at the end of W2015 for the first time since 2002. In the density range 32.36 – 32.39
kg m$^{-3}$, these layers remained thicker than ~450 m during W2016 to W2018. This indicates low



stratification at intermediate depths and a favorable preconditioning of intermediate waters for deep
convection due to W2015 deep convection.
B(zi) is our estimate of the preconditioning of the water column before winter (see Method). Fig. 6a
shows that, deeper than 100 m, B was smaller for W2016, W2017 and W2018 than for W2015 or for
the mean W2008 − W2014. Furthermore, for W2016, W2017 and 2018, B remained nearly constant
with depth between 600 and 1,300 m, which means that once the water column has been
homogenized down to 600 m, little additional buoyancy loss results in homogenization of the water
column down to 1,300 m. Both conditions (i) less buoyancy to be removed and (ii) absence of
gradient in the B profile down to 1,300 m indicate a more favorable preconditioning of the water
column for W2016, W2017 and W2018 than during W2008 − W2015.
To understand the relative contributions of $\theta$ and S to the preconditioning, we computed the thermal
($B_\theta$) and haline ($B_s$) components of B (B = $B_\theta$ + $B_s$). In general, $B_\theta$ ($B_s$) increases with depth when $\theta$
decreases (S increases) with depth. On the contrary, a negative slope in $B_\theta$ ($B_s$) corresponds to $\theta$
increasing (S decreasing) with depth and is indicative of a destabilizing effect. The negative slopes in
$B_\theta$ and $B_s$ profiles are not observed simultaneously because density profiles are stable.
We describe the relative contributions of $B_\theta$ and $B_s$ to B by looking first at the mean 2008 − 2014
profiles (discontinuous blue lines in Fig. 6). $B_\theta$ accounts for most of the increase in B from the surface
to 800 m and below 1,400 m. The negative slope in the $B_s$ profile between 800 − 1,000 slightly
reduces B and is due to the decrease in S associated with the core of LSW (see Fig. 3 in Piron et al.
2016). In the layer 1,000 − 1,400 m, the increase in B is mainly explained by $B_s$, which follows the
increase in S in the transition from LSW to Iceland Scotland Overflow Water (ISOW) that will be
referred to hereinafter as the deep halocline. The evaluation of the preconditioning of the water
column was usually analyzed in terms of heat (e.g., Piron et al. 2015; 2017). The decomposition of B
in $B_\theta$ and $B_s$ reveals that $\theta$ governs B in the layer 0 − 800 m. S tends to reduce the stabilizing effect of
$\theta$ in the layer 800 − 1,000 m, and reinforces it in the layer 1,000 − 1,400 m in adding up to $1 \times 10^9$ J m$^2$
to B.
In order to further understand why the Irminger Sea was favorably preconditioned during winters
2016 − 2018, we compare the $B_\theta$ and $B_s$ of W2017, which was the most favorably preconditioned
winter, with the mean 2008 − 2014 (Fig. 7a). From the surface to 1,600 m, $B_\theta$ and $B_s$ were smaller for
W2017 than for the mean 2008 − 2014. There are two additional remarkable features. First, in the
layer 500 − 1000 m, the large reduction of $B_\theta$, in relation to its mean 2008 − 2014, mostly explains the
decrease of B in this layer. Second, the more negative value of $B_s$ in the layer 1,100 − 1,300 m,
compared to its mean 2008 − 2014, eroded the $B_\theta$ slope, making the B profile more vertical for



W2017 than in the mean. The more negative contribution of $B_s$ in the layer 1,100 – 1,300 m is related
to the fact that the deep halocline was deeper for W2017 (1,300 m, see red discontinuous line in Fig.
7a) than for the mean 2008 – 2014 (1,000 m, see blue discontinuous line in Fig. 7a). Finally, we note
that the profiles of $B(z_i)$, $B_\theta(z_i)$ and $B_s(z_i)$ for W2016 and W2018 are more similar to the profiles of
W2017 than to those of W2015 or to the mean 2008 – 2014 (see Fig. 6), which indicates that the
water column was also favorably preconditioned for deep convection in W2016 and W2018 for the
same reasons than in W2017.
The origin of the changes in B is now discussed from the evolution of the monthly anomalies of θ, S
and $\sigma_0$ at a point in the Irminger Sea (59°N – 40°W, Fig. 8). These anomalies were computed using
ISAS (Gaillard et al., 2016) and were referenced to the monthly mean of 2002 – 2016. A positive
anomaly of $\sigma_0$ appeared in 2015 between the surface and 600 m (Fig. 8a) and reached 1,100m in
2016 and beyond. This positive anomaly of $\sigma_0$ correlates with a negative anomaly of θ that, however,
reached ~1,400 m depth in 2016 that is deeper than the positive anomaly of $\sigma_0$. The negative
anomaly of S between 1,000 - 1,500 m that started in 2016 caused the negative anomaly in $\sigma_0$
between 1,200 – 1,500 m (the negative anomaly in θ between 1,200 – 1,400 m does not balance the
negative anomaly of S).
The θ and S anomalies in the water column during 2016 – 2018 explain the anomalies of B, $B_\theta$ and $B_s$
and can be summarized as follows. On the one hand, the properties of the surface waters (down to
500 m) were colder than previous years and, despite they were also fresher, they were denser. The
density increase in the surface water reduced the density difference with the deeper-lying waters.
The intermediate layer (500 – 1000 m) was also favorably preconditioned due to the observed
cooling. Additionally, in the layer 1,100 – 1,300 m, the large negative contribution of $B_s$ in relation to
its mean is explained by the decrease in S in this layer, which caused a decrease in $\sigma_0$ and,
consequently, reduced the $\sigma_0$ difference with the shallower-lying water. The decrease in S also
resulted in a deepening of the deep halocline.

## 4.4. Atmospheric forcing versus preconditioning of the water column

We now use the estimates of atmospheric forcing ($B_{surf}$* + $BF_{ek}$) to predict the maximum convection
depth for a given winter based on September profiles of B. The predicted convection depths are
determined as the depth at which $B(zi)$ equals the atmospheric forcing. The associated error was
estimated considering the error in the atmospheric forcing ($0.05 \times 10^9$ J m$^{-2}$). We found predicted
convection depths of 1175 ± 10 m, 1270 ± 25 m, 1425 ± 10 m and 1285 ± 20 m for W2015, W2016,
W2018, respectively. The Q3 estimated from W2016 and W2017 observations (1,325 m



and 1,400 m, respectively) are very close to the predicted convection depth. In W2015, the predicted
convection depth was underestimated compared to the observed Q3 (1,310 m). The contrary is
observed for W2018; this result is in line with the fact that Q3 in W2018 was most likely
underestimated since 6 out of the 10 profiles from which deep convection was recorded dived down
to 1,100 m, which coincided with their MLD. When these 6 profiles are excluded of the analysis, Q3 is
1,300 m, which is within the error bar of the predicted convection depth.
The satisfactory predictability of the convection depth validates our neglect of the horizontal
advection and indicates that deep convection occurred locally. Finally, this demonstrates that in spite
the atmospheric forcing was close to mean conditions during W2016, W2017 and W2018, convection
depths > 1300 m were reached, which was only possible thanks to the favorable preconditioning.

## 5. DISCUSSION
Deep convection happens in the Irminger Sea during specific winters because of strong atmospheric
forcing (high buoyancy loss), favorable preconditioning (low stratification) or both at the same time
(Pickart et al., 2003). Strong atmospheric forcing explained for instance the very deep convection
(reaching depth greater than 1500 m) observed in the early 90s (Pickart, et al., 2003) and in W2015
(de Jong et al. 2016; Piron et al. 2017), and the return of deep convection after many years without
convection in W2008 (Väge et al., 2009) and in W2012 (Piron et al., 2016). The favorable
preconditioning caused by the densification of the convected layer at the end of W2008 favored a
new deep convection event in W2009 despite neutral atmospheric forcing (de Jong et al. 2012).
Similarly, the preconditioning observed after W2015 favored deep convection in W2016 (this work).
Our study reveals that the preconditioning surprisingly persisted along three consecutive winters
(2016 – 2018) which allowed deep convection although atmospheric forcing was close to the
climatological values. Why did this favorable preconditioning persist in time?
The favorable preconditioning of the water column during 2016 – 2018 in the Irminger Sea resulted
from two hydrological anomalies affecting different ranges of the water column: the cooling of the
layer 200 – 800 m and the freshening of 1,200 – 1,400 m layer. Note that, the cooling affected the
layer surface – 1,400 m and the freshening affected the layer 1,000 – 1,500 m (Fig. 8), but the θS
anomalies were density compensated in the layer 1,000 – 1,200 m.
We see in Fig. 5a a sudden decrease in θ in the intermediate layers compared to the previous years.
It indicates that the decrease in θ of the layer 200 – 800 m likely originated locally during W2015
when extraordinary deep convection happened. The freshening of the layer 1,200 – 1,400 m
appeared in 2016 (Fig. 8c). Given its depth range, it is unlikely that this anomaly was locally formed.





Moreover, this anomaly is different to that affecting the intermediate layer because density
increased in the intermediate layer with respect to the mean 2002 – 2016, while it decreased in the
1,200 – 1,400 m layer (Fig. 8a). Our hypothesis is that the S anomaly originated in the Labrador Sea
and was further transferred to the Irminger Sea by the cyclonic circulation encompassing the
Labrador Sea and Irminger Sea at these depths (Daniault et al., 2016; Ollitrault & Colin de Verdière,
2014). It is corroborated by the 2D evolution of the anomalies in S in the layer 1,200 – 1,400 m (Fig.
9): a negative anomaly in S appeared in the Labrador Sea in February 2015, which was transferred
southward and northeastward in February 2016 and intensified over the whole SPNA in February

386    2017.

We now compare the atmospheric forcing and the preconditioning of the water column in the
Irminger Sea with those of the nearby Labrador Sea where deep convection happens each year. As
noted by Pickart et al. (2003), the atmospheric forcing over the Labrador Sea is ~15 % larger than that
over the Irminger Sea: the means (1993 - 2018) of the atmospheric forcing, defined as the time -
accumulated $B_{surf}$* + $BF_{ek}$ from 1 September to 31 March the year after, are -2.61 ± 0.55 x $10^9$ J $m^{-2}$ in
the Labrador Sea and -2.26 ± 0.58 x $10^9$ J $m^{-2}$ in the Irminger Sea. The difference was larger during the
period 2016 – 2018 when the atmospheric forcing equaled -3.10 ± 0.19 x $10^9$ J $m^{-2}$ in the Labrador
Sea and -2.31 ± 0.21 x $10^9$ J $m^{-2}$ in the Irminger Sea. In terms of preconditioning, the 2008 – 2014
mean B profile (blue continuous lines in Fig. 7) was lower by ~0.5 x $10^9$ J $m^{-2}$ in the Labrador Sea than
in the Irminger Sea for the surface to 1,000 m layer and by more than 1 x $10^9$ J $m^{-2}$ below 1,000 m. It
indicates that the water column was more favorably preconditioned in the Labrador Sea than in the
Irminger Sea. Differently, B for W2017 shows slightly lower values from the surface to 1,300 m in the
Irminger Sea than in the Labrador Sea (see orange lines in Fig. 7). However, B in the Labrador Sea
remains constant down to the depth of the deep halocline between LSW and North Atlantic Deep
Water (NADW) at 1,700 m. In the Irminger Sea, the deep halocline remained at ~1,300 m between
2016 and 2018 (see $B_s$ lines in Fig. 7a). Differently, in the Labrador Sea, the deep halocline was
successive deepening from 1,200 m for the mean to 1,735 m, 1,775 m and 1905 m in W2016, W2017
and W2018, respectively (see discontinuous lines in Fig. 7b). The deep halocline acts as a physical
barrier for deep convection in both the Irminger Sea and the Labrador Sea, but because it is deeper
in the Labrador Sea than in the Irminger Sea, a deeper convection depth is granted in the former
than in the latter. Summarizing, the atmospheric forcing and the preconditioning of the water
column are in general more favorable for deep convection in the Labrador Sea than in the Irminger
Sea. In winters 2016 - 2018 in the Labrador Sea, both atmospheric forcing and preconditioning of the
water column favored by a deeper than average deep halocline, granted the deepest convection
depth ever observed in the Labrador Sea (comparison of our results with those of Yashayaev and





Loader, 2017). Contrasting, in the Irminger Sea, during the same period, the atmospheric forcing was
close to climatological values, and the favorable preconditioning of the water column allowed 1,300
m depth convection, what was exceptional for the Irminger Sea.
In the following we consider the time – evolution of $\theta$, S and $\sigma_0$ at the layer 700 – 900 m (Fig. 10),
considered here as the core of the LSW in both the Irminger Sea and the Labrador Sea. From 2002 to
2012, a progressive increase in the $\theta$ and S of the LSW core in the Irminger Sea is noticeable despite
the high frequency variability. The $\theta$ and S changes were not density compensated causing a decadal
decrease in   $\sigma_0$   of ~0.01 Kg m$^{-3}$. From 2012 to 2015, both $\theta$ and S decreased, while $\sigma_0$ remained
constant. During 2015 – 2018, $\theta$ decreased from 3.6 °C to 3.2 °C, and $\sigma_0$ increased from 27.72 to
27.75 kg m$^{-3}$ (see also Fig. 3 and Table 1). In spite of the cooling and densification that occurred
during the last winters, LSW is warmer and lighter than that formed during W1994 and W1995
(2.85°C, 27.78 kg m$^{-3}$, Pickart et al. 2003). We note a long-term (1994 – 2018) warming of LSW
observed in the Irminger Sea. The comparison with the LSW properties in the Labrador Sea over 2002
– 2018 (Fig. 9) shows that the LSW observed in both basin has the same density while that of the
Irminger Sea is warmer and saltier than that of the Labrador Sea. Interestingly, this behaviour was
also observed along the 90s (Pickart et al, 2003). It is also worth noting that $\theta$ and S observed in
Labrador Sea and Irminger Sea converged at the end of the 90s (Fig. 6 in Pickart et al., 2003) and
along our period 2015 – 2018 (Fig. 3 and Fig. 10). However, there is an imporant difference between
the two periods: deep convection was not observed in the Irminger Sea at the end of the 90s while it
was very intense during 2015 – 2018. This disparity might indicate that Labrador Sea and Irminger
Sea are evolving differentely and only further observations would disclose the origin and mechanisms
causing the differences.
Climate models forecast increasing input of freshwater in the North Atlantic due to ice-melting under
present climate change (Bramber et al., 2018), which could reduce, or even shut-down, the deep
convection in the North Atlantic (Yang et al., 2016; Brodeau & Koenigk, 2016). We observed a fresh
anomaly in the surface waters in regions close to the eastern coast of Greenland in 2016 that
extended to the whole Irminger Sea in 2017 (Fig. S4). However, at the moment, the surface
freshening did not hamper the deep convection in the Irminger Sea possibly because the surface
water also cooled, which favors the preconditioning for deep convection. Swingedouw et al., 2013
indicated that the freshwater signal due to Greenland ice sheet melting is mainly accumulating in the
Labrador Sea. However, no negative anomaly of S was detected in the surface waters of the Labrador
Sea (Fig. S4). It might be explained by the intense deep convection affecting the Labrador Sea since
2014 that could have transferred the surface freshwater anomaly to the ocean interior. This suggest

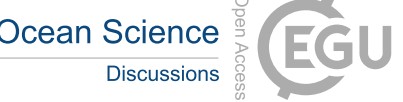



that, in the last years, the interactions between expected climate change anomalies and the natural
dynamics of the system combined to favor very deep convection. This however does not foretell the
long term response to climate change.

## 6. CONCLUSIONS

During 2015 – 2018 winter deep convection happened in the Irminger Sea reaching deeper than
1,300 m. It is the first time deep convection was observed in the Irminger Sea during four
consecutive winters. LSW formed in the Irminger Sea from 2015 to 2018 get colder, fresher and
denser, being similar in 2018 to the properties of the LSW formed in the Labrador Sea.
Considering the expected increase in freshwater inputs, the atmospheric forcing and preconditioning
of the water column was evaluated in terms of buoyancy. We showed that the atmospheric forcing is
10% weaker when evaluated in terms of buoyancy than in terms of heat because of the non-
negligible effect of the freshwater flux. The analysis of the preconditioning of the water column in
terms of buoyancy to be removed (B) and its thermal and salinity terms ($B_\theta$ and $B_s$) revealed that $B_\theta$
dominated the B profile from the surface to 800 m and $B_s$ reduced the B in the 800 – 1000 m layer
because of low salinity of LSW. Deeper, $B_s$ increased B due to the deep halocline (LSW-ISOW) that
acted as a physical barrier limiting the depth of the convection.
During 2016 – 2018, the air-sea buoyancy losses were close to the climatological values and the very
deep convection was possible thanks to the favorable preconditioning of the water column. It was
surprising that these events reached convection depths similar to those observed in W2012 and
W2015, when the latter were provoked by high air-sea buoyancy loss intensified by the effect of
strong wind stress. It was also surprising that the water column remained favorably preconditioned
during three consecutive winters without strong atmospheric forcing. In this paper, we studied the
reasons why this happened.
The favorable preconditioning for deep convection during 2016 – 2018 was due to the combination
of two types of hydrological anomalies affecting different depth ranges. First, the intermediate water
(200 – 800 m) was favorably preconditioned because buoyancy (density) decreased (increased) due
to the cooling caused by the deep convection of W2015. Second, buoyancy (density) increased
(decreased) in the layer 1,200 – 1,400 m due to the decrease in S caused by the lateral advection of
fresher LSW formed in the Labrador Sea. The S anomaly of this layer resulted in a deeper deep
halocline. Hence, the cooling of the intermediate water was essential to reach convection depth of



800 – 1,000 m, and the freshening in the layer 1,200 – 1,400 m and the associated deepening of the
deep halocline, allowed the very deep convection (> 1,300 m) in W2016 – W2018.
**Author contribution**: PZ treated and analyzed the data. PZ and HM interpreted the results. PZ, HM
and VT discussed the results and wrote the paper.

**ACKNOWLODGEMENT**
The Argo data were collected and made freely available by the International Argo Program and the
national programs that contribute to
it. (http://www.argo.ucsd.edu, http://argo.jcommops.org). The Argo Program is part of the Global
Ocean Observing System. The NAO data were downloaded from the UCAR Climate Data Guide
website (Schneider et al., 2013): https://climatedataguide.ucar.edu/climate-data/hurrell-north-
atlantic-oscillation-nao-index-pc-based. The Ssalto/Duacs altimeter products were produced and
distributed by the Copernicus Marine and Environment Monitoring Service (CMEMS)
(http://www.marine.copernicus.eu).

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





Table 1. Properties of the deep convection in the Irminger Sea and in the Labrador Sea in winters
2015 – 2018. We show: the maximal MLD observed, the aggregate maximum depth of convection
Q3, the $\sigma_0$, $\theta$ and S of the winter mixed layer formed during the convection event and n, which is the
number of Argo profiles indicating deep convection. The uncertainties given with $\sigma_0$, $\theta$ and S are the
standard deviation of the n values considered to estimate the mean values.

| IRMINGER SEA | Maximal MLD | Q3 MLD | $\sigma_0$ | $\theta$ | Salinity | n |
|---|---|---|---|---|---|---|
| W2015 | 1715 | 1310 | 27.732 ± 0.007 | 3.494 ± 0.139 | 34.868 ± 0.015 | 37 |
| W2016 | 1575 | 1325 | 27.745 ± 0.004 | 3.444 ± 0.150 | 34.877 ±0.017 | 7 |
| W2017 | 1400 | 1400 | 27.746 ± 0.006 | 3.324 ± 0.113 | 34.864 ± 0.009 | 4 |
| W2018 | 1300 | 1100/1300* | 27.751 ± 0.007 | 3.188 ± 0.058 | 34.854 ± 0.013 | 10 |
| | | | | | | |
| LABRADOR SEA | Maximal MLD | Q3 MLD | $\sigma_0$ | $\theta$ | Salinity | n |
| W2015 | 1675 | 1504 | 27.733 ± 0.009 | 3.279 ± 0.036 | 34.842 ± 0.010 | 41 |
| W2016 | 1801 | 1620 | 27.743 ± 0.006 | 3.124 ± 0.047 | 34.836 ± 0.010 | 18 |
| W2017 | 1780 | 1674 | 27.752 ± 0.008 | 3.172 ± 0.029 | 34.853 ± 0.009 | 26 |
| W2018 | 2020 | 1866 | 27.756 ± 0.006 | 3.145 ± 0.083 | 34.855 ± 0.010 | 13 |

*Q3 estimated when the data of Float 5903102 were excluded of the analysis. We exclude them
because their MLDs matched with the maximal depth dived by the float.







**FIGURES**

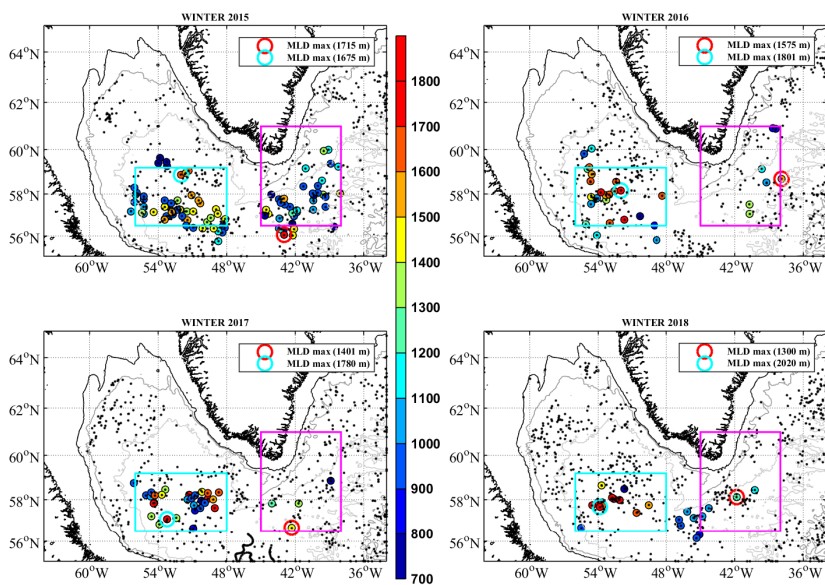

**Figure 1.** Position of all Argo floats north of 55°N in the Atlantic between 1 January and 30 April a)
2015, b) 2016, c) 2017 and d) 2018 (small black points). The colored big points and colorbar indicate
the depth of the mixed layer depth (MLD) when MLD deeper than 700 m, which indicates deep
convection. The circles correspond to the positions of the profiles with the maximum MLD for the
given box. The pink and cyan boxes delimit the regions used for estimating the time series of
atmospheric forcing and the vertical profiles of buoyancy to be removed in the Irminger Sea and
Labrador Sea, respectively (Irminger Sea: 56.5°N − 61.0°N and 45.0°W − 38.0°W, Labrador Sea:
56.5°N – 59.2°N and 56°W – 48°W).





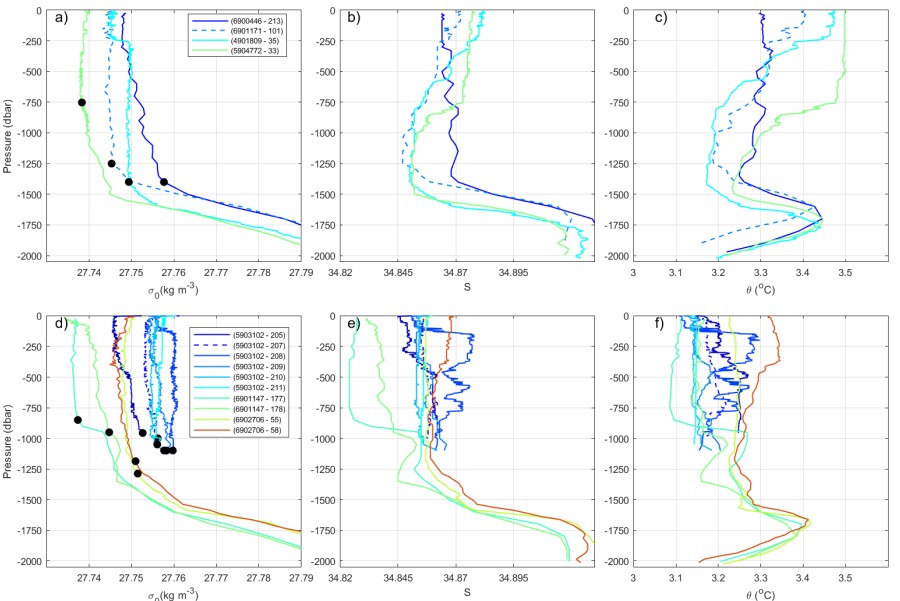


**Figure 2**. Vertical distribution of $\sigma_0$, S and $\theta$ of Argo profiles showing MLD deeper than 700 m in the Irminger Sea in Winter 2017 (a, b and c) and in Winter 2018 (d, e, f). The black points indicate the MLD in each profile. In the legend, the float and cycle of each profile are indicated.


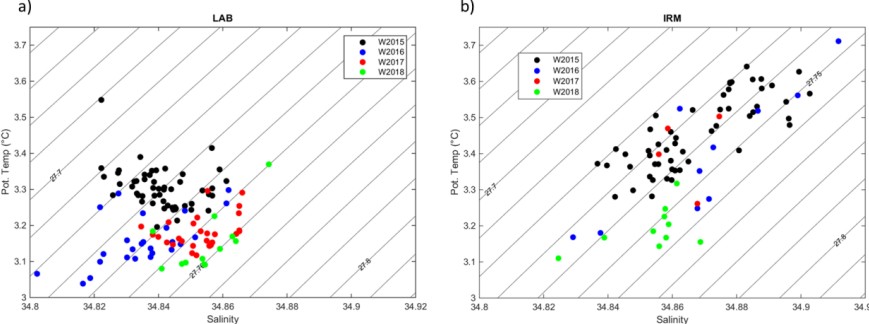

**Figure 3.** Diagram TS of the n profiles with MLD deeper than 700 m found a) in the Labrador Sea and
b) in Irminger Sea, in the winters 2015, 2016, 2017 and 2018. The properties of each profile were
estimated as the vertical mean between 200 m and the MLD.

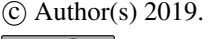


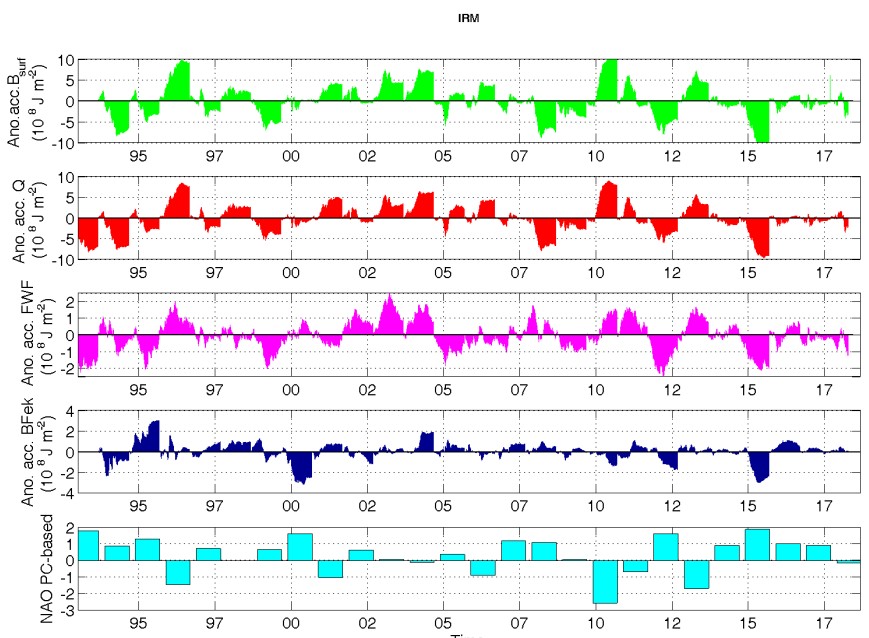

**Figure 4.** Time series of anomalies of accumulated Bsurf*, Q, FWF* and BF$_{ek}$, averaged in the Irminger
Sea. They are anomalies with respect to 1993 – 2016. The accumulation was from 1 September to 31
August the following year. The winter NAO index (Hurrel et al., 2018) is also represented in the
bottom panel.



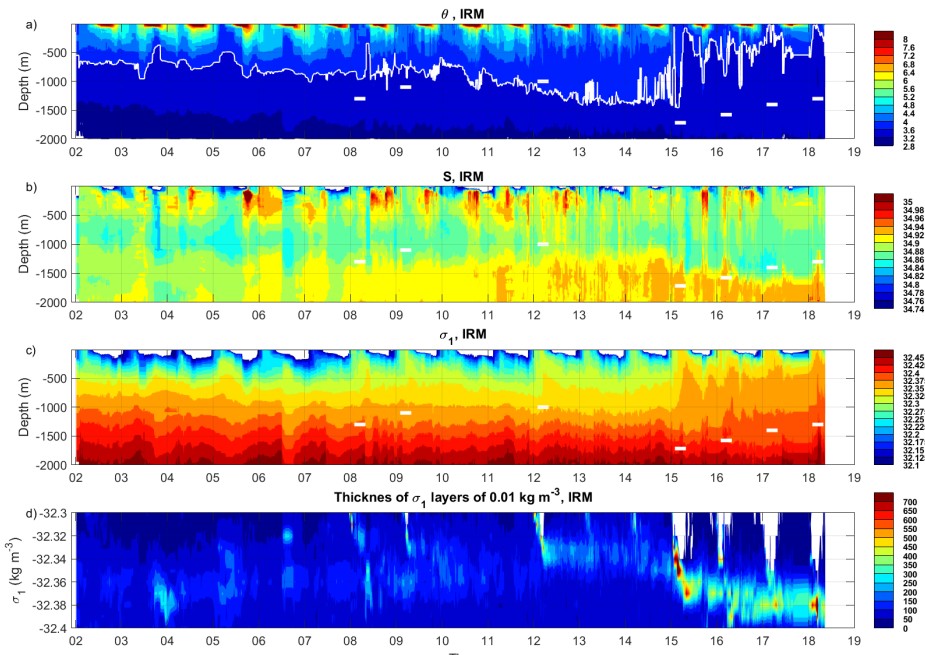

**Figure 5.** Time-evolution of vertical profiles measured from Argo floats in the Irminger Sea: a) θ ; b) S; c) $\sigma_1$ and d) thickness of 0.01 kg m$^{-3}$ thick $\sigma_1$ layers. The white horizontal bars in plots a), b) and c) indicate the maximal convection depth observed in the Irminger Sea when deep convection occurred. The white line in plot a) indicates the depth of the isotherm 3.6 °C. These figures were created from all Argo profiles reaching deeper than 1000 m in the IRM region (56.5° – 61°N, 45°– 38°W, pink box in Fig. 1). The yearly numbers of Argo profiles used in this figure are shown in Fig. S1.



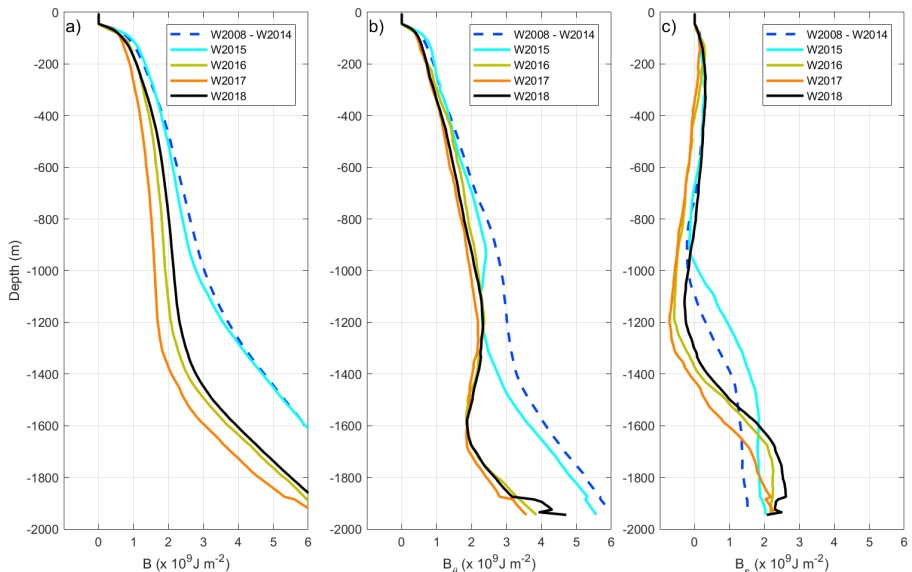

637

**Figure 6.** Vertical profile of a) the total buoyancy to be removed (B), b) the thermal component ($B_\theta$)

and c) the salinity component ($B_s$). They were calculated from Argo data measured in the Irminger

box (see Fig. 1) in September before the winter indicated in the legend. For W2018, we considered

data from 15/08/2017 to 30/09/2017 because not enough data were available in September 2017.

The number of Argo profiles taken into account to estimate the B profiles was more than ten for all

the winters.



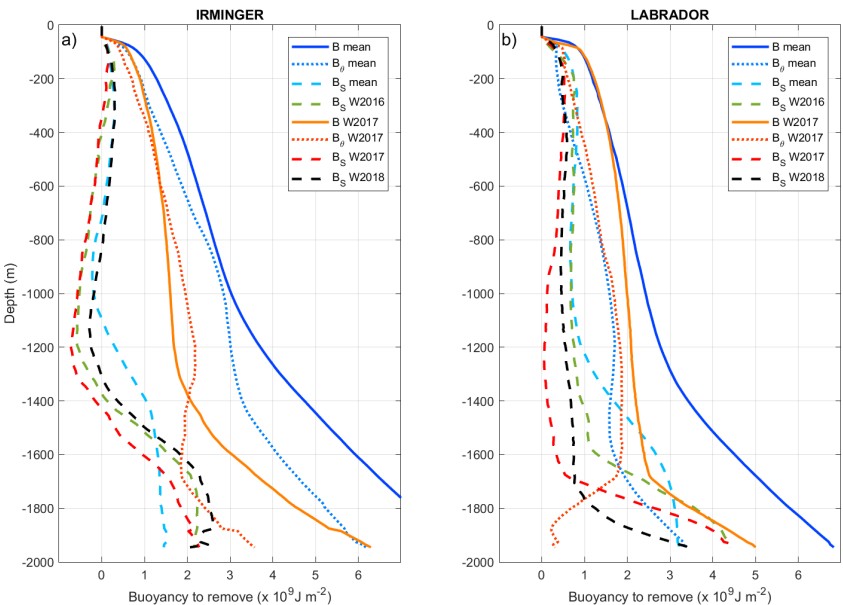

644

Figure 7. Decomposition of profiles of Buoyancy to be removed (B, continuous lines) in its thermal
($B_\theta$, point lines) and salinity ($B_s$, discontinuous lines) components in a) the Irminger Sea; b) the
Labrador Sea. To compare the mean 2008 − 2014 with W2017 compare reddish lines with bluish
lines. The $B_s$ component in W2016 and W2018 was added in order to show the evolution of the
depth of the deep halocline in both the Irminger Sea and the Labrador Sea.





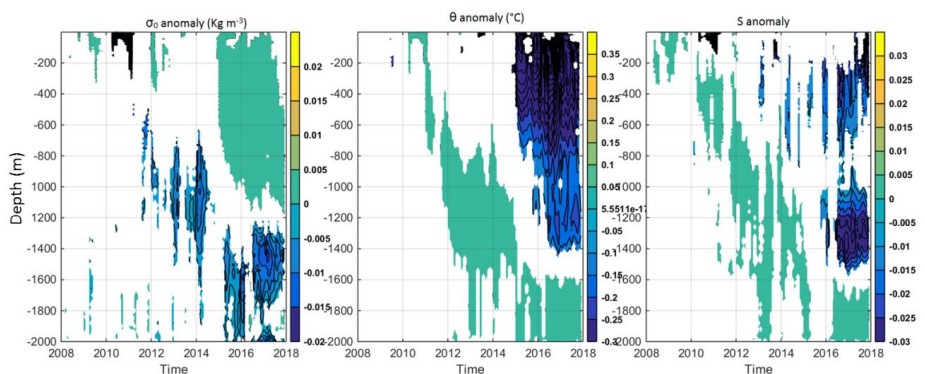

**Figure 8.** Evolution of vertical profiles of monthly anomalies of $\sigma_0$ (left panel), $\theta$ (central panel) and S
(right panel), at 59°N, 40°W in the Irminger Sea. The anomalies were estimated from the ISAS
database (Gaillard et al., 2016), they were referenced to the monthly mean estimated for 2002 –
2016. We represented only anomalies larger than one standard deviation of the mean and since
2008 in order to see clearly the recent changes. All anomalies are displayed in Fig. S3.

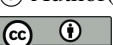



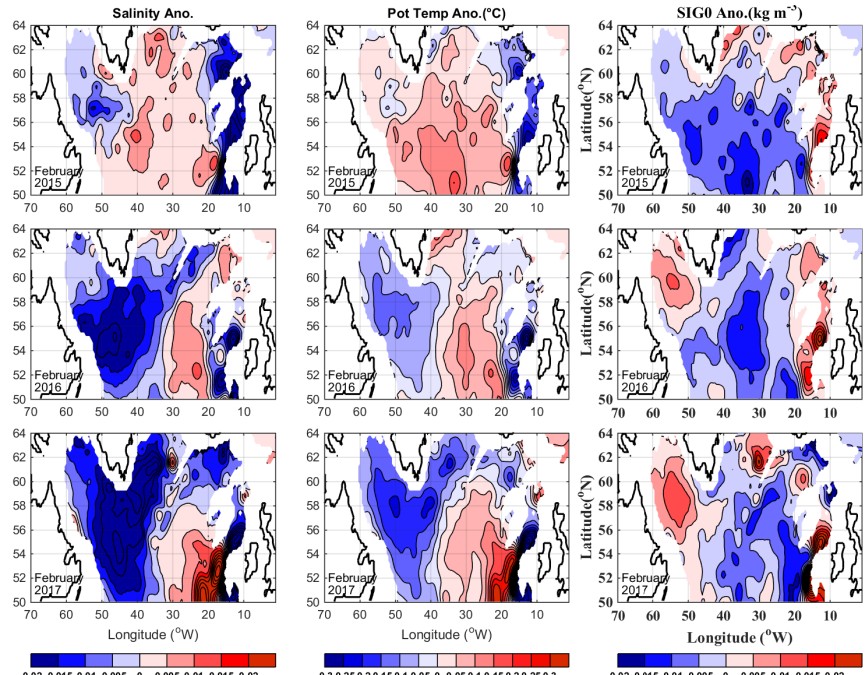

**Figure 9.** Horizontal distribution of the anomalies of S (left panels), θ (central panels) and σ₀ (right
panels) in the layer 1200 – 1400 m in February 2015 (upper panels), February 2016 (central panels)
and February 2017 (lower panels). The monthly anomalies were estimated from ISAS database
referenced to the period 2002 – 2016.

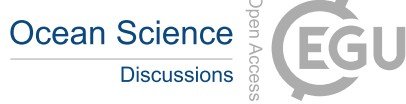

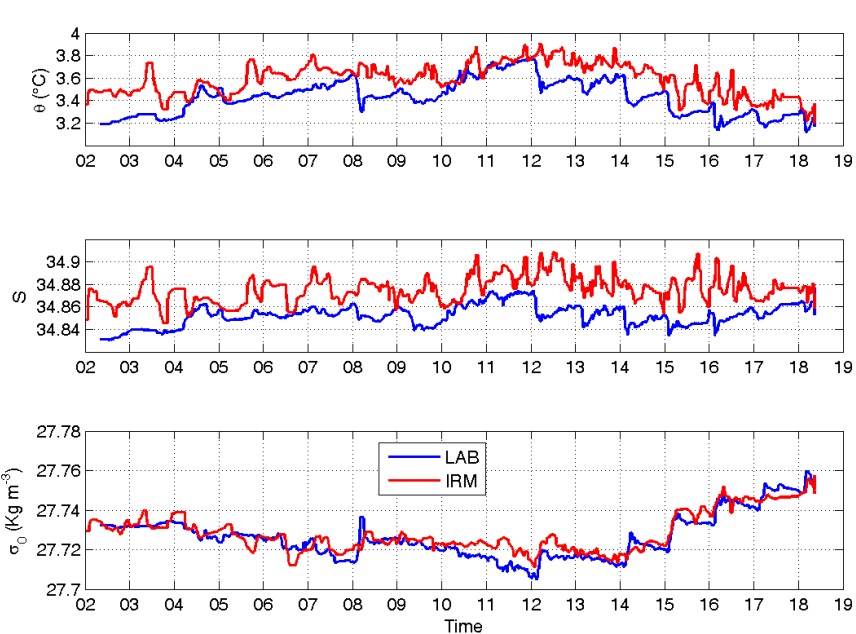


Figure 10. Time-evolution of the properties of the LSW core (700 – 900 m) in the Irminger Sea (red) and Labrador Sea (blue), estimated from all Argo data in the pink and cyan boxes in Fig. 1.





