# Peer review of "Why did deep convection persist over four consecutive winters (2015-2018) Southeast of Cape Farewell?"

_Ocean Science, 2019_

## Referee Comment (RC1) · Femke de Jong (Referee) · 21 Jun 2019

Review of: "Two superimposed cold and fresh anomalies enhanced Irminger Sea deep convection in 2016-2018" by Zunino, Mercier and Thierry

The manuscript is interesting to read and a nice update on the latest convective activity in the western subpolar gyre. The separation of buoyancy fluxes into different components, including those from Ekman transport, is interesting. However it's not too surprising to see that the Ekman contribution is small given that the horizontal SST gradients are also relative small. Overall I would like to see this paper published eventually, but there is at least one major issue that need to be addressed before.

The paper hangs on the derivation of mixed layer depths and the comparison with

previous years published in literature. This comparison is currently troublesome because of the substantially different way the authors derive/define the mixed layer. In fact, some of the derived mixed layers depths do not appear to be associated with actual/recent mixed layers. Although some of the results may be robust to the methods, some others (e.g. max depth per winter, match with predicted MLDs) will clearly have to be adjusted. This should be addressed before publication.

More specifically, in a layer with turbulent mixing all properties, density, salinity and temperature, are homogenized. If the mixing occurred very recently (on the order of days ago), the homogeneous profile will still be visible all the way down to the bottom of the mixed layer. In the literature that is referred to for previous mixed layer depths (de Jong et al., 2012, 2018; de Jong in de Steur, 2016), we therefore always specified that all three properties should be mixed and of the bottoms of the mixed layer identified in each property we take the shallowest as the final mixed layer. Similar criteria were applied by group of Vage et al. in their papers. In the mooring data, as well as Argo, there are cases at the end of the winter where there remains no steplike feature visible in the density profiles at all and where a density criterion would strongly overestimate mixing, while such as steplike feature always remains visible in T and S. Therefore it is even more important to take all variables into account.

The difference between this definition of a mixed layer and that of the authors, which is a density-only criterion, is especially clear in Figure 2. The top three panels show density, salinity and potential temperature profiles from the winter of 2017. The bright blue profile, which the authors identified as having the deepest mixed layer, appears to be somewhat mixed in T and S in the upper 250 dbar (though even that is a bit questionable) but it is clearly stratified between 250 and 1400 dbar. In fact, the stratification in temperature is quite large ($\sim$0.25°C) for the Irminger Sea. The only profile in the set of four that could (potentially) qualify as having a mixed layer is the greenish profile. This would nearly half the winter maximum mixed layer depth and may also affect how well the predicted MLD match the observations.

There is code readily available to derive MLD from Argo profiles using all variables (Holte and Talley, 2009; http://mixedlayer.ucsd.edu/). I suggest the authors use this, or some adjustment of their own code, to rederive the MLD for all profiles and adjust the results of the paper accordingly.

My final main comment is that the title could be rephrased to represent the content/conclusions better. The fresh anomaly that seems to be referred to a deep one, the lowering of the halocline. The surface freshwater anomaly, which is discussed in detail elsewhere but is only touched upon here, is was not enhancing convection. It is only the cold surface anomaly that worked to enhance somewhat, but even that is only touched upon. Still, those who have not yet read the abstract may think this paper is about the big surface Sanom currently going around. While in fact, the paper focuses in detail on favorable preconditioning which is not mentioned in the title. So, it is not clear why this title was chosen.

Below are some more minor comments

Introduction

Line 94. "In the Labrador Sea, deep convection occurs almost every year, yet with different intensity. In the Irminger Sea...". In the Irminger Sea some convection ($\sim$400 m) always occurs as well, and the intensity varies not unlike the Labrador Sea. Please rephrase or add a definition of "deep".

Data

Why is the TEOS-10 toolbox used, but profiles of theta and practical salinity are still shown instead of CT and SA?

Please explain briefly why 35 is chosen as a reference.

The ERA Interim reanalysis is replaced by ERA5. Best to do a check whether the results are robust to the choice of reanalysis.

[Figure]

Method

De Boyer and Montégut criterion is not suitable for these profiles as discussed above.

The definition of the Irminger Sea, with 48°W as the limit is rather unusual. The area in Figure 1 southeast of Cape Farewell is not typically referred to as the Irminger Sea as it fall outside of the central Irminger Gyre and profiles here are very likely to have been recently advected from the Labrador Sea. To be more consistent with previous literature it would be better to split this region in three areas: the Labrador Sea, the Irminger Sea and in between the area south of Cape Farewell.

Equation 1 and others. There are periods (.) instead of multiplication symbols.

Results

What is Q3?

Part of the results paragraph will have to be rewritten when MLD are rederived.

Line 268: Mean over which period?

Line 296: This is true only when the upper 600 m already has a density close to that of the layer below (which for example could not be the case when a lot of freshwater is added). Otherwise additional buoyancy fluxes will still be required.

Section 4.4

It would be good to compare fluxes closer to the position of the observed deep MLs. These are sometimes on the very boundary of the box used to calculate the winter flux.

The method used to predict the MLD does not take advection into account. This is counterintuitive because we see advection play a big role throughout winter in the field. The fact that the reanalysis do not quite match with the actual fluxes observed at OOI (Josey et al, 2018) may also be needed to take into account here. It will be interesting to see how much of a match between prediction and observation remains once new

MLD are derived, likely the prediction will overestimate more.

Discussion

Line 366: This was seen throughout the 1990s and is not quite as surprising as the authors state.

Line 397: The Labrador Sea is always more favorably preconditioned, it is quite visible in the hydrographic sections and has been noted before.

Line 406: Bit of a chicken and egg problem. The halocline is also deeper in the Labrador Sea because convection is deeper there. Would rephrase.

Line 416 / Fig 10. The depth is chosen such that it is always in the convective regime in the Labrador Sea, hence the nice steps. It is mostly too deep for this in the Irminger Sea, so a lot of the variability is caused by advection except in exceptionally deep convection years.

Line 430: There is a multitude of evidence that there was very deep convection in the Irminger Sea in the 1990s (but no Argo program). The LSW was advected to the Irminger Sea in the subsequent years and hence properties converged. Please rephrase.

Line 435: Bamber et al?

Please reference papers by Dukhovsky et al (2019) and Holliday et al (2019) who both describe the fresh anomaly.

Conclusions

Line 450 "in or near the Irminger Sea"

Line 473: was this only caused by advection of LSW or was the layer eroded by the 1600 m deep convection in 2016?

---

## Referee Comment (RC2) · Anonymous Referee #2 · 25 Jun 2019

This paper reports on a very interesting analysis of recent Argo data in the subpolar North Atlantic. They claim that deep convection in the Irminger Sea, which began in 2015, persisted through 2018 because of favorable preconditioning. They perform some novel analyses and the findings will be of great interest to the community. However, I agree with the review posted by Femke de Jong, which raises issues with the way that mixed layers are defined in the study. This is central to the interpretation and conclusions of the study, and I think that at the very least some major re-framing of the work is necessary. I recommend this work for publication after major revisions.

Major comments

I would like to echo de Jong's comments regarding the mixed layer depth derivation. In order to show that deep convection occurred in 2016-2018, they should show that all

properties (including temperature and salinity) were homogeneous throughout, not just that a density threshold was exceeded at a very deep depth. Regardless of the method selected by the authors in their revision they should include much more detail on it in the text as it is a central calculation. They should also be clear about how sensitive their results are to the method used and to the thresholds that are selected. They should also detail how their methods relate to the methods used by previous studies in the region.

The authors should address how sensitive their results are to the Argo float coverage, and what portion of the Argo floats present have deep mixed layers. They report how many floats have mixed layers deeper than 700m, but not how many were present. Does the percentage of floats with deep mixed layers decrease over time? The authors should comment on why they think so few Argo floats have deep mixed layers. Is it consistent with their buoyancy forcing analysis? Does the sampling in time account for some of this: i.e. Are deep mixed layers seen more commonly late in winter?

The mixed layers reported in winter 2018 are almost all to the south of Cape Farewell, and not in the Irminger Sea. Further, the TS properties in 2018 are much more similar to Labrador Sea properties than Irminger Sea properties (Figure 3). This is consistent with the SCF box properties reported in Piron et al. 2017. Some of the properties in 2016 and 2017 may also fall in that category, I don't think the author's should be calling this "Irminger Sea convection".

The author's show a very interesting analysis of Labrador Sea properties which are advected into Irminger Sea and contribute to the deepening of the Irminger Sea halocline (Figures 7 and 9). Is this advection limited to the 1200-1400 range they consider? How does advection from the Labrador Sea in other depth ranges fit in?

The title is confusing, and I wonder if, in general, the author's should shift their focus from the Irminger Sea in particular and instead focus on the important connections between the intermediate waters in the subpolar North Atlantic. I think the authors have

an opportunity here to clarify that intermediate waters are formed in many places and how the connections between these basins affect intermediate water mass properties. Their focus on salinity in addition to temperature would make this angle particularly interesting.

Minor comments

The link between anthropogenic forcing and the recent convection that is drawn in the first few sentences in the abstract and throughout the introduction is a bit of a stretch. The motivation could be made more direct and convincing, and this type of speculation could remain in the discussion where it is more relevant.

L109: "for the first time to our knowledge in this region" - this is a broad claim and not necessary.

Section 3.1: Please add significant detail on the mixed layer estimation method.

L222: The 2018 profiles with deep mixed layers are not in the Irminger Sea.

L237: "Water masses formed are very similar" It should at least be acknowledged that they are formed much closer to the Labrador Sea than in previous years.

L247: maybe instead: "heat alone" at the end of this sentence.

L248: This paragraph is confusing. Perhaps referring to Figure 4 earlier on would help?

L331: "despite they were also fresher" → "despite the fact that they were also fresher"

L340: Refer to figure 6.

L348: Clarify what happened here. These floats only profiled down to 1,100m?

L351: I was also confused by the fact that the author's claim to neglect advection, but cite advection of properties from the Labrador Sea as a reason for favorable preconditioning. Perhaps remove that claim. Additionally, the fact that the T and S properties are not homogeneous goes against the idea that deep convection is occurring locally.

L370: hydrological → hydrographic

L370: anomalies relative to what?

L383: Why would only the properties in the 1200-1400 depth range be advected? Or are they the only ones that have a profound effect? See above. Please clarify.

L415/Figure 10: Not sure how this figure and paragraph are linked to the rest of the study.

L470: hydrological → hydrographic

Figure 4: Note the differences between the axis ranges in the caption. This figure could be featured earlier as it provides important context.

Figure 5: From Figure 5d, it appears that the thick density layers are actually below the densities that are being ventilated in the Irminger Sea (white areas). This supports the idea that they are being advected from the Labrador Sea.

Figure 6: Please clarify: are you using all Argo data within the box, or only the ones with deep mixed layers?

Figure 7: This is a very interesting figure! Could feature more prominently and be used to describe some key differences between the Labrador and Irminger Seas. Reddish → red. Bluish → blue.

Figure 8: missing a) b) c) labels on the figure.

---

## Referee Comment (RC3) · Anonymous Referee #3 · 26 Jun 2019

**Two superimposed cold and fresh anomalies enhanced Irminger Sea deep convection in 2016 - 2018**

by Patricia Zunino, Herlé Mercier, and Virginie Thierry

In this manuscript persistence of deep convection in the Irminger Sea is investigated. One winter of particularly severe atmospheric forcing and deep convection was followed by three winters of climatological strength which also had deep mixed layers. The authors quantified the buoyancy loss required for deep convection to commence each winter and concluded that the preconditioning arising from the previous winter's homogenization of the water column was a main reason for the persistence of deep convection.

I think this manuscript has the potential to be an important and valuable contribution to better understand deep water formation in the Irminger Sea /subpolar North Atlantic. However, as made clear also by the other reviewers, I have concerns about the determination of mixed-layer depths. As such, I recommend that the paper be revised before publication.

**Major comments:**

I am not convinced that automated routines, such as the threshold or split and merge methods, are particularly suitable for determining the vertical extent of the mixed layer. These routines generally perform well when applied to summer and fall profiles, when the upper ocean is stratified and there is a pronounced density difference between the mixed layer and the lower part of the profile. However, they are less accurate during periods of active convection when stratification is eroded. Furthermore, such routines cannot identify mixed layers that are isolated from the surface, either in the form of vertically stacked mixed layers or by early stages of surface restratification. Such isolated mixed layers are prevalent in the Labrador and Irminger Seas during winter (e.g. Pickart *et al.*, 2002). As pointed out by the other referees, if the density profile is considered in isolation, changes in temperature and salinity may be density-compensated such that the water column can appear to be homogenized while in reality it is not. Examples of that can be seen in Figure 2a-c (in particular 4901809 - 35). To avoid erroneous mixed-layer depths, I strongly recommend employing the semi-objective method developed by Pickart *et al.* (2002) instead of relying on automated routines.

Deep convection evidently took place in winter 2015 as documented by the many deep mixed layers shown in Figure 1. For winters 2016-2018, on the other hand, the vast majority of the Irminger Sea profiles do not have particularly deep mixed layers. If widespread deep convection occurred also during these winters, there should be many more profiles with deep mixed layers. Is it possible that the mixed-layer depths determined by the automated routines are remnants of deep convection from a previous winter or from the Labrador Sea where mixed layers are generally deeper? To get a more robust estimate of convection in the subpolar North Atlantic these winters, I suggest dispensing with the 700 m "deep convection" criterion and showing if not every mixed layer at least the 50-80% deepest mixed layers encountered by each float every winter. That would remove shallow mixed layers arising from early phases of the seasonal evolution of the mixed layer and profiles obtained within stratified eddies, while the remaining mixed layers would allow for more robust quantification of the general depth of convection.

Profiles that do not extend beneath the base of the mixed layer (there may be some examples in Figure 2d-f) would result in a shallow bias of the mixed-layer depth estimate and should be excluded from the analysis.

**Specific comments:**

Line 95:
It should be: "...Argo and **mooring** data..."

Lines 106 and 361:
Mixed layers exceeding 1400 m depth were determined also from shipboard measurements in the Irminger Sea in April 2015 (Fröb *et al.*, 2016).

Line 122:
If the TEOS-10 convention is used, conservative temperature and absolute salinity should be used instead of potential temperature and salinity.

Line 123:
Please explain why a salinity of 35 was chosen as a reference value.

Line 124:
Please provide more information about the gridded products. Are different time periods and resolutions the only difference between the products? What are the errors, in particular for the EN4 product which extends back to 1900 and covers some very data-sparse periods?

Line 130:
Does the net air-sea heat flux include radiative fluxes or only turbulent fluxes?

Line 149:
I do not think that 48°W is commonly used as a border between the Labrador and the Irminger Seas. Many of the deep mixed layers were recorded directly south of Greenland, in a region that is not really part of either the Labrador or the Irminger Seas.

Line 156 and elsewhere:
Please insure that all papers cited in the text are included in the References section. For example is Gill (1982) missing.

Line 174:
How was the depth of the Ekman layer estimated?

Line 179:
For consistency, it might be better to use SST also from the EN4 product.

Line 185:
It should be: "...most of **the** Argo profiles..."

Line 197:
It should be: "...to be removed (B(zi)) **from** the late summer density profile..."

Line 234:
Salted, in this context, is not appropriate. "Became saltier" would be a better expression.

Line 284:
If B remained nearly constant, does that imply that restratification and advection are unimportant?

Line 297:
Units (m) are missing after 800-1000.

Line 321:
What was the basis for choosing the point 59°N, 40°W?

Line 377:
If convection exceeded 1400 m in winter 2014-15 (e.g. Fröb *et al.*, 2016), why is it unlikely that this layer was locally formed?

Line 382:
The papers by Lavender *et al.* (2000) and Straneo *et al.* (2003) could also be cited here.

Line 383:
Corroborated is misspelled.

Line 388:
If deep convection occurs every year, perhaps the definition of deep convection should be revised.

Line 403:
It should be: "...the deep halocline was **successively** deepening..."

Line 410:
I am sceptical of the claim that the deepest convection-depth ever observed in the Labrador Sea occurred in winters 2016-2018. Very likely convection in the successive high-NAO winters of the early 1990s substantially exceeded convection in winters 2016-2018. At that time mixed-layer depths were at least 2300 m (e.g. Avsic *et al.*, 2006).

Lines 419 and 421:
Density units are not capitalized consistently.

Line 425:
It should be: "...observed in both **basins**..."

Line 430:
There were no wintertime measurements in the Irminger Sea in the early 1990s, but there is strong indirect evidence that deep convection occurred in the Irminger Sea at that time (see for example publications from the group of R. Pickart).

Line 481:
Acknowledgement is misspelled.

Lines 519 and 522:
The name de Jong is inconsistently capitalized.

Figure 5:
Please indicate, for example using tick marks along the top axis, when Argo float profiles were available in the Irminger Sea.

**References**

Avsic T, Karstensen J, Send U, Fischer J. 2006. Interannual variability of newly formed Labrador Sea Water from 1994 to 2005. *Geophysical Research Letters* **33**: L21S02, doi:10.1029/2006GL026 913.

Fröb F, Olsen A, Våge K, Moore G, Yashayaev I, Jeansson E, Rajasakaren B. 2016. Irminger Sea deep convection injects oxygen and anthropogenic carbon to the ocean interior. *Nature Communications* **7**: doi:10.1038/ncomms13 244.

Lavender KL, Davis RE, Owens WB. 2000. Mid-depth recirculation observed in the interior Labrador and Irminger Seas by direct velocity measurements. *Nature* **407**: 66–69.

Pickart RS, Torres DJ, Clarke RA. 2002. Hydrography of the Labrador Sea during active convection. *Journal of Physical Oceanography* **32**: 428–457.

Straneo F, Pickart RS, Lavender KL. 2003. Spreading of Labrador Sea Water: An advective-diffusive study based on Lagrangian data. *Deep Sea Research I* **50**: 701–719.

---

## Author Comment (AC1) · 7 Oct 2019

Review of: "Two superimposed cold and fresh anomalies enhanced Irminger Sea deep convection in 2016-2018" by Zunino, Mercier and Thierry

The manuscript is interesting to read and a nice update on the latest convective activity in the western subpolar gyre. The separation of buoyancy fluxes into different components, including those from Ekman transport, is interesting. However it's not too surprising to see that the Ekman contribution is small given that the horizontal SST gradients are also relative small. Overall I would like to see this paper published eventually, but there is at least one major issue that need to be addressed before.

The paper hangs on the derivation of mixed layer depths and the comparison with previous years published in literature. This comparison is currently troublesome because of the substantially different way the authors derive/define the mixed layer. In fact, some of the derived mixed layers depths do not appear to be associated with actual/ recent mixed layers. Although some of the results may be robust to the methods, some others (e.g. max depth per winter, match with predicted MLDs) will clearly have to be adjusted. This should be addressed before publication.

More specifically, in a layer with turbulent mixing all properties, density, salinity and temperature, are homogenized. If the mixing occurred very recently (on the order of days ago), the homogeneous profile will still be visible all the way down to the bottom of the mixed layer. In the literature that is referred to for previous mixed layer depths (de Jong et al., 2012, 2018; de Jong in de Steur, 2016), we therefore always specified that all three properties should be mixed and of the bottoms of the mixed layer identified in each property we take the shallowest as the final mixed layer. Similar criteria were applied by group of Vage et al. in their papers. In the mooring data, as well as Argo, there are cases at the end of the winter where there remains no steplike feature visible in the density profiles at all and where a density criterion would strongly overestimate mixing, while such as steplike feature always remains visible in T and S. Therefore it is even more important to take all variables into account.

The difference between this definition of a mixed layer and that of the authors, which is a density-only criterion, is especially clear in Figure 2. The top three panels show density, salinity and potential temperature profiles from the winter of 2017. The bright blue profile, which the authors identified as having the deepest mixed layer, appears to be somewhat mixed in T and S in the upper 250 dbar (though even that is a bit questionable) but it is clearly stratified between 250 and 1400 dbar. In fact, the stratification in temperature is quite large (_0.25_C) for the Irminger Sea. The only profile in the set of four that could (potentially) qualify as having a mixed layer is the greenish profile. This would nearly half the winter maximum mixed layer depth and may also affect how well the predicted MLD match the observations.

There is code readily available to derive MLD from Argo profiles using all variables (Holte and Talley, 2009; http://mixedlayer.ucsd.edu/). I suggest the authors use this, or some adjustment of their own code, to rederive the MLD for all profiles and adjust the results of the paper accordingly.

My final main comment is that the title could be rephrased to represent the content/conclusions better. The fresh anomaly that seems to be referred to a deep one, the lowering of the halocline. The surface freshwater anomaly, which is discussed in detail elsewhere but is only touched upon here, is was not enhancing convection. It is only the cold surface anomaly that worked to enhance somewhat, but even that is only touched upon. Still, those who have not yet read the abstract may think this paper is about the big surface Sanom currently going around. While in fact, the paper focuses in detail on favorable preconditioning which is not mentioned in the title. So, it is not clear why this title was chosen.

Thank you very much for your constructive comments. In the following we answer point by point to your comments and indicate how the manuscript is going to be revised.

Following your suggestion, we revised the manuscript to define the MLDs based on density, temperature and salinity criteria (and not density criteria only). We adapted our method to include temperature and salinity criteria in addition to density criteria and we compared our results to two alternative methods of determination of the MLD previously used by de Jong et al. (2012) and Pickart et al. (2002). In our revised method, we determined the MLD as the shallowest of the three MLD estimates obtained separately from temperature, salinity and density profiles using the threshold method (de Boyer Montégut et al., 2004). The threshold criteria were the differences in property between the surface (30 m) and the MLD set to 0.01 kg m$^{-3}$ in density (Piron et al. 2017), 0.1°C in temperature and 0.012 in salinity. The temperature threshold of 0.1°C and the salinity threshold of 0.012 were selected because they correspond to a threshold of 0.01 kg m$^{-3}$ in density that was previously shown to perform well in the subpolar gyre (Piron et al., 2016). Indeed, MLD based on this density threshold favorably compared to those estimated by the method of Thomson and Fine (2003) as demonstrated in Piron et al. (2016; 2017) and visual inspection.

We used de Jong's methodology as follows. First we interpolated the Argo data into 10 m depth steps. Then, we estimated the standard deviations of density, temperature and salinity from the surface to each depth level. Following de Jong et al. method's, three MLD were defined as the depths were the standard deviations were smaller than 0.05 kg m$^{-3}$, 0.05°C and 0.005 for density, temperature and salinity, respectively. The final MLD was the shallowest of the three estimates.

The Pickart's methodology was applied as follows. We used the estimates of our threshold method as a first guess for the MLD. Then, the mean and standard deviation of the density, temperature and salinity were estimated from the surface to the initially defined MLD. Finally, we plotted the two–standard deviation envelope overlaid on the original profile. The mixed layer depth was determined as the location where the profile permanently crossed outside of the two–standard deviation envelope.

The MLDs resulting from our method are shallower than the MLD resulting from the method of de Jong et al. (see examples in figures R1 – R3). Moreover, sometimes, the MLD defined by de Jong's method in terms of temperature or salinity is not placed at the base of the mixed layer (as visually defined), e.g. profiles 6900446 – 213 (Fig. R1) or 5904772 – 33 (Fig. R3). Otherwise, the MLDs

estimated by our method are coherent with the MLDs resulting from the method of Pickart et al. (2002): see the envelopes (discontinuous vertical lines in figures R1 – R3) of mean ± two - times the standard deviation of density, salinity and potential temperature, from the surface to the MLD estimated with our method. Finally, we also compared our results with the MLDs determined using Holte & Talley (2009)'s method and available in the web. However, MLDs were not available for all our floats, e.g. float 6900446, or the method provides too shallow MLD, e.g. profile 6901171 – 101 (89 m, see Fig. R2).

[Figure]

Figure R1. Vertical profiles of potential density, salinity and potential temperature of profile 6900446 - 213. The black points are the MLD estimated by our threshold method. The blue points indicate the MLDs resulting from the method of de Jong et al. (2012): in the density plot the MLD derived from density profile, in the salinity plot the MLD derived from salinity profile and in the temperature plot the MLD derived from temperature profile; the final MLD is the shallowest of the three defined MLDs. Following Pickart et al. (2002), the envelopes of mean ± two - times the standard deviation of the density, salinity and potential temperature from the surface to the MLD estimated using as a first guess for the MLD our threshold method were estimated and represented as discontinuous vertical lines.

[Figure]

Figure R2. Same than Fig. R2 but for profile 6901171 – 101. Additionally, the horizontal lines on the left side plot represent the MLDs estimated by the Holte and Talley's method: in gray the MLD defined by the density threshold, in black the MLD defined by the density algorithm, in blue the MLD defined by the temperature threshold and in cyan the MLD defined by the temperature algorithm; note that black and blue lines are overlapping.

[Figure]

Figure R3. Same than Fig. R2 but for profile 5904772 - 33.

Because the comment of the referee focused on profiles for winter 2017, we expose in more details here the differences between the previous and the new MLD estimates for this winter 2017. First, the profile 4901809 – 35 has been eliminated because the stratification of the upper 250 m corresponds to the seasonal stratification (this profile was measured on 29[th] April 2017). In any case, applying the criterion of temperature threshold of 0.1°C, the MLD would be 337 m, shallower than 700 m. Second, the MLD of the profile 6901171 – 101 changes from 1250 m (the previous estimate) to 801 m (the new estimate). The MLDs estimated for profiles 6900446 – 213 and 5904772 – 33 do not change.

The MLDs of all the profiles measured Southeast Cape Farewell (SECP) during winters 2015 – 2018 were recalculated with our revised method. The positions and MLDs of the profiles showing MLDs deeper than 700 m are represented in Fig. R4. Comparing these new results with the previous results, we find that the number of profiles showing MLD deeper than 700 m decreased: 31 profiles (new) in place of 36 (previous) for winter 2015, 3 profiles (new) in place of 7 profiles (previous) for winter 2016, 3 profiles (new) in place of 4 profiles (previous) for winter 2017 and 9 profiles (new) in place of 10 profiles (previous) for winter 2018.

We have also recalculated all the properties showed in the table 1 of the previous version of the paper. Note that these properties are now estimated considering only the profiles inside the SECF box (pink box in Fig. R4.) The new results (table R1 in this document) are in line with the results of the submitted paper.

Table R1. Properties of the deep convection in the SECF (56.5°N-59.3°N, 45°W – 38°W) in winters 2015 – 2018. We show: the maximal MLD observed, the aggregate maximum depth of convection Q3, the $\sigma_0$, $\theta$ and S of the winter mixed layer formed during the convection event and n, which is the number of Argo profiles indicating deep convection. The uncertainties given with $\sigma_0$, $\theta$ and S are the standard deviation of the n values considered to estimate the mean values.

| | Maximal MLD (m) | Q3 (m) | $\sigma_0$ (Kg m$^{-3}$) | $\theta$ (°C) | S | N |
|---|---|---|---|---|---|---|
| W2015 | 1710 | 1205 | 27.733 ± 0.007 | 3.478 ± 0.130 | 34.866± 0.013 | 29 |
| W2016 | 1575 | 1471 | 27.746± 0.002 | 3.388 ± 0.032 | 34.871± 0.003 | 3 |
| W2017 | 1400 | 1251 | 27.745± 0.007 | 34.868± 0.007 | 3.364± 0.109 | 3 |
| *W2018 | *1300 | *1250 | *27.752± 0.004 | *34.857± 0.003 | *3.204± 0.069 | *4 |
| W2018 | 1300 | 1300 | 27.748± 0.001 | 34.859± 0.003 | 3.263± 0.031 | 2 |

*W2018 line corresponds to the properties of the mixed layer in W2018 in SEFC when the data of Float 5903102 were considered in the analysis. Finally, following the suggestion of referee 3, we decide to exclude the data of float 5903102 of our analysis because their MLDs matched with the maximal depth dived by the float.

[Figure]

Figure R4. Positions of all Argo float north of 55°N in the Atlantic between 1 January and 30 April a) 2015, b) 2016, c) 2017 and d) 2018 (black and colored points). The colored points and color bar indicate the mixed layer depth (MLD) when MLD was deeper than 700 m. The pink circles indicate the position of the maximal MLD observed SECF each winter. The pink and cyan boxes delimit the regions used for estimating the time series of atmospheric forcing and the vertical profiles of buoyancy to be removed in the SECF region and the Labrador Sea, respectively (SECF: 56.5°N – 59.3°N and 45.0°W – 38.0°W, Labrador Sea: 56.5°N – 59.2°N and 56°W – 48°W).

We want also to clarify that in the previous version of the paper and in the new results, the deepest MLD observed in the SECF in winter 2017 was recorded by profile 6900446 – 213 and not by profile 4901809 – 35 (bright blue profile in the Fig. 2 of the previous version of the paper) as indicated by the referee. Note that for 6900446 – 213, the new MLD is the same than in the previous version of the manuscript.

Concluding, when recalculating the MLDs as suggested by the referees, the maximal MLD observed in the SECF was deeper than 1300 m in winters 2016, 2017 and 2018 (see fig. R4 and table R1). It indicates that deep convection occurred during the studied winters. This is the first important result of our paper, which does not change when recalculating the MLD.

Concerning the title, in order to avoid preconceived ideas to the reader, in the revised manuscript we change it to:
"Why did convection persist over 4 consecutive winters (2015-2018) South East of Cape Farewell?"

Below are some more minor comments
Introduction
Line 94. "In the Labrador Sea, deep convection occurs almost every year, yet with different intensity. In the Irminger Sea…". In the Irminger Sea some convection (_400 m) always occurs as well, and the intensity varies not unlike the Labrador Sea. Please rephrase or add a definition of "deep".
We agree. Following Piron et al. (2015), we focus on convection deeper than 700 m, which is the minimum MLD for LSW renewal. We clarified the sentence that now reads :
"In the Irminger Sea, Argo and mooring data showed that  convection **deeper than 700 m** happened  during winters 2008, 2009, 2012, 2015 and 2016 (…)."

Data
Why is the TEOS-10 toolbox used, but profiles of theta and practical salinity are still shown instead of CT and SA?
TEOS-10 allows the computation of theta and practical salinity.

Please explain briefly why 35 is chosen as a reference.
This sentence is going to be deleted because we do not use FW in the paper. Sorry for the confusion it may have caused.

The ERA Interim reanalysis is replaced by ERA5. Best to do a check whether the results are robust to the choice of reanalysis.
It could be interesting to check the results obtained using the new ERA5 dataset. However, the first author of this paper, who processed the data, is now working in a private company and she has not the time of redoing calculations with this new database.

Method
De Boyer and Montégut criterion is not suitable for these profiles as discussed above.
See above our answer to the major comment.

The definition of the Irminger Sea, with 48_W as the limit is rather unusual. The area in Figure 1 southeast of Cape Farewell is not typically referred to as the Irminger Sea as it fall outside of the central Irminger Gyre and profiles here are very likely to have been recently advected from the Labrador Sea. To be more consistent with previous literature it would be better to split this region in three areas: the Labrador Sea, the Irminger Sea and in between the area south of Cape Farewell.

We agree that 48° W is not the limit between Labrador and Irminger Sea. When splitting the region in three areas as in Piron et al. (2017) we did not observe deep convection in the northernmost Irminger Sea (note that with the previous method of MLD computation we had a few deep MLD in the northernmost Irminger Sea in winter 2016 (those MLD corresponding to profiles not homogenous in temperature and salinity were not diagnosed with the new MLD method). In the new version of the paper we define a new pink box that we refer to as Southeast Cape Farewell (SECF) region (see Figure R4). The only change in the pink box is its northern limit: 61°N/59.3°N in the previous/revised version of the manuscript. The new box encloses all the profiles showing deep MLD during winter 2016, 2017 and 2018 Southeast of Cape Farewell. Note that the pink box is also used to estimate the atmospheric forcing and the preconditioning of the region. We recalculated it: the new results are very similar to the results shown in the previous version of the paper and do not change the conclusions of the paper.

Equation 1 and others. There are periods (.) instead of multiplication symbols. Thank you for noting it. We change all of them.

Results
What is Q3? "$Q_3$ is the MLD value that is exceeded by 25% of the profiles showing MLD deeper than 700 m and is equivalent to the aggregate maximum depth of convection defined by Yashayaev and Loder (2016).", as it was indicated in lines 152 – 153 of the submitted manuscript.

Part of the results paragraph will have to be rewritten when MLD are rederived.
Right, we are going to rewrite this section with the new results.

Line 268: Mean over which period?
1993 – 2016, as indicated in the figure caption of Figure 4. We add 1993 – 2016 in the text.

Line 296: This is true only when the upper 600 m already has a density close to that of the layer below (which for example could not be the case when a lot of freshwater is added). Otherwise additional buoyancy fluxes will still be required.

We are describing the buoyancy profiles from the mean (2008 – 2014) and we see that the thermal component of the buoyancy dominates the total buoyancy. We agree that if a large amount of freshwater is added to the upper ocean, we would find an important contribution of the haline component of the buoyancy, but it is not what we see in the mean (2008 – 2014) buoyancy profiles. We added Fig. 6 at the end of this sentence to make clear that we are describing the results of this figure and that the statement is not a general statement.

Section 4.4
It would be good to compare fluxes closer to the position of the observed deep MLs. These are sometimes on the very boundary of the box used to calculate the winter flux.

This comment has also motivated us to reduce the SECF or pink box. The new estimates of atmospheric forcing correspond to a reduced region closer to the position where deep convection took place.

The method used to predict the MLD does not take advection into account. This is counterintuitive because we see advection play a big role throughout winter in the field. The fact that the reanalysis do not quite match with the actual fluxes observed at OOI (Josey et al, 2018) may also be needed to take into account here. It will be interesting to see how much of a match between prediction and observation remains once new MLD are derived, likely the prediction will overestimate more.

Your comment makes sense, but note that the new estimates of MLD continue matching adequately with the predicted MLD. In the new version of the manuscript we will mention that the differences between the predicted and observed convection depth could be due to errors in the atmospheric forcing (Josey et al., 2018), lateral advection and/or spatial variation in the convection intensity within the box that was not captured by the Argo sampling.

Discussion
Line 366: This was seen throughout the 1990s and is not quite as surprising as the authors state.
We deleted "surprisingly".

Line 397: The Labrador Sea is always more favorably preconditioned, it is quite visible in the hydrographic sections and has been noted before.
The Labrador Sea is *usually* more favorably preconditioned than the Irminger Sea. However, we see that the water column from the surface to 1,300 m in winter 2017 is more favorably preconditioned in the SECF than in the Labrador Sea (see Fig. 7 in the previous version). For example, in order to homogenize the water column down to 1,300 m, $1.80 \times 10^9$ J m$^{-2}$ is required in the SECF whereas $2.13 \times 10^9$ J m$^{-2}$ is needed in the Labrador Sea.

Line 406: Bit of a chicken and egg problem. The halocline is also deeper in the Labrador Sea because convection is deeper there. Would rephrase.
Not really a chicken and egg problem, if you are thinking in terms of preconditioning. To clarify our point we modified the sentence as : "The deep halocline acts as a physical barrier for deep convection in both the Irminger Sea and the Labrador Sea, but because the deep halocline is deeper in the Labrador Sea than in the Irminger Sea, the preconditioning is more favorable to deeper convection in the Labrador Sea than in the Irminger Sea."

Line 416 / Fig 10. The depth is chosen such that it is always in the convective regime in the Labrador Sea, hence the nice steps. It is mostly too deep for this in the Irminger Sea, so a lot of the variability is caused by advection except in exceptionally deep convection years.
You are right and the figure is confusing even when the discussion is limited to deep convection events in the SECF region. Because of your comment and the comments of reviewer 2 we decide to delete Figure 10 and paragraph 415 - 433 in the revised manuscript.

Line 430: There is a multitude of evidence that there was very deep convection in the Irminger Sea in the 1990s (but no Argo program). The LSW was advected to the Irminger Sea in the subsequent years and hence properties converged. Please rephrase.

We decide to remove Figure 10 and paragraph 415 -433 in the revised manuscript. It does not change the conclusions of the paper.

Line 435: Bamber et al? Yes, Bamber et al. Thank you for noticing it.

Please reference papers by Dukhovsky et al (2019) and Holliday et al (2019) who both describe the fresh anomaly.
Dukhovsly et al. (2019) describe the freshwater anomaly of the 2010s, so, it does not concern the period we study in our paper.
Otherwise, we think that Holliday et al (2019) has not been published yet (V. Thierry is co-author of the paper).

Conclusions
Line 450 "in or near the Irminger Sea"
In the revised manuscript this sentence is changed to:
"During 2015 – 208 winter deep convection happened in SECF reaching deeper than 1,300 m".

Line 473: was this only caused by advection of LSW or was the layer eroded by the 1600 m deep convection in 2016?
Our sentence was confusing. We will mention that deep convection of W2016 also favored the preconditioning for winter 2017 – 2018.

---

## Author Comment (AC2) · 7 Oct 2019

This paper reports on a very interesting analysis of recent Argo data in the subpolar North Atlantic. They claim that deep convection in the Irminger Sea, which began in 2015, persisted through 2018 because of favorable preconditioning. They perform some novel analyses and the findings will be of great interest to the community. However, I agree with the review posted by Femke de Jong, which raises issues with the way that mixed layers are defined in the study. This is central to the interpretation and conclusions of the study, and I think that at the very least some major re-framing of the work is necessary. I recommend this work for publication after major revisions.

Thank you very much for your constructive review. In the following we answer to each of your comments and describe how we are going to take into account your suggestions in the revised manuscript.

Major comments

I would like to echo de Jong's comments regarding the mixed layer depth derivation. In order to show that deep convection occurred in 2016-2018, they should show that all properties (including temperature and salinity) were homogeneous throughout, not just that a density threshold was exceeded at a very deep depth.

The three referees agreed on this point. Consequently, we have adapted our methodology to estimate the MLD considering density, temperature and salinity profiles. Please, refer to the beginning of the answer to de Jong (referee 1) in order to see how we have modified our methodology to estimate MLD. The new results (MLD and properties) do not change the main conclusions of our paper.

Regardless of the method selected by the authors in their revision they should include much more detail on it in the text as it is a central calculation. They should also be clear about how sensitive their results are to the method used and to the thresholds that are selected. They should also detail how their methods relate to the methods used by previous studies in the region.

Right, we will explain our revised methodology to estimate the MLD indicating the threshold of density, temperature and salinity used. Moreover, we will add a figure in supplementary material showing that our estimates of MLD favorable compare with the estimates resulted when using the methods of Pickart et al. (2002) or de Jong et al. (2012) as discussed at the beginning of response to referee 1.

The authors should address how sensitive their results are to the Argo float coverage, and what portion of the Argo floats present have deep mixed layers. They report how many floats have mixed layers deeper than 700m, but not how many were present. Does the percentage of floats with deep mixed layers decrease over time? The authors should comment on why they think so few Argo floats have deep mixed layers. Is it consistent with their buoyancy forcing analysis? Does the sampling in time account for some of this: i.e. Are deep mixed layers seen more commonly late in winter?

Thanks to your comment we realized that our discussion was misleading because it was based on the percentage of profiles showing deep convection during the *entire* winter, which is small by construction because only profiles at the end of winter show deep convection. We rather should have count the number of floats showing deep convection during a given year. Accordingly, we now identify the period when deep convection occurs as the period when at least one profile shows MLD > 700 m (the period begins when a profile with MLD > 700 m is detected for the first time for the given winter and it ends when there is no more profiles with MLD > 700 m). Then, we quantified the

percentage of floats with deep MLD present in that period and region (pink box in figure R4 of answer to referee 1). This information is summarized in table R2 of this document and will be included in section 4.1 of the revised manuscript. The percentage varies between 33% and 73%. In 2017, the three profiles with deep mixed layer were recorded by three different floats, all located in the southwest corner of our region. This shows that the convection area was confined to a small area of the SECF region and explains that the lowest percentage is observed in 2017.

Table R2. Sensitivity study about the Argo float coverage in the SECF region (pink box in Figure R4 of the answer to referee 1). Period is the period during which floats with deep mixed layers were observed. We indicate the total number of floats found in the SECF region during the indicated period, and the number of floats showing deep convection. Finally, the percentage of floats showing deep convection is indicated.

| | Period | n floats in the region | n floats in the region with deep convection | percentage of floats in the region with deep convection |
|---|---|---|---|---|
| W2015 | 15/01/2015 to 21/04/2015 | 11 | 8 | 73% |
| W2016 | 22/02/2016 to 21/03/2016 | 4 | 2 | 50% |
| W2017 | 16/03/2017 to 04/04/2017 | 9 | 3 | 33% |
| W2018 | 24/02/2018 to 26/03/2018 | 4 | 2 | 50% |

The mixed layers reported in winter 2018 are almost all to the south of Cape Farewell, and not in the Irminger Sea. Further, the TS properties in 2018 are much more similar to Labrador Sea properties than Irminger Sea properties (Figure 3). This is consistent with the SCF box properties reported in Piron et al. 2017. Some of the properties in 2016 and 2017 may also fall in that category, I don't think the author's should be calling this "Irminger Sea convection".
Right. In the revised manuscript we changed the northern limit of the pink box to 59.3°N instead of 61°N previously and refer to the pink box as Southeast Cape Farewell (SECF).

The author's show a very interesting analysis of Labrador Sea properties which are advected into Irminger Sea and contribute to the deepening of the Irminger Sea halocline (Figures 7 and 9). Is this advection limited to the 1200-1400 range they consider? How does advection from the Labrador Sea in other depth ranges fit in?
Advection from the Labrador Sea certainly contributed to vary the properties from the surface to 1000 m. However, the buoyancy budget showed that this is minor contribution compared to the buoyancy loss due to the local air-sea flux. We add a comment about it in the revised manuscript.

The title is confusing, and I wonder if, in general, the author's should shift their focus from the Irminger Sea in particular and instead focus on the important connections between the intermediate waters in the subpolar North Atlantic. I think the authors have an opportunity here to clarify that intermediate waters are formed in many places and how the connections between these basins

affect intermediate water mass properties. Their focus on salinity in addition to temperature would make this angle particularly interesting.

In the revised manuscript we change the title to: "Why did convection persist over 4 consecutive winters (2015-2018) South East of Cape Farewell?"

Moreover, we now mention several times the role of advection from the Labrador Sea. We also added to the discussion the following paragraph:

The Labrador Sea, SECF region and Irminger Sea are three distinct deep convection sites (e.g. Yashayaev et al., 2007; Bacon et al., 2003; Pickart et al., 2003; Piron et al., 2017). In this work, we give new insights on the connections between the different sites, showing how lateral advection of fresh LSW formed in the Labrador Sea favored the preconditioning in the SECF region fostering deeper convection."

Minor comments

The link between anthropogenic forcing and the recent convection that is drawn in the first few sentences in the abstract and throughout the introduction is a bit of a stretch. The motivation could be made more direct and convincing, and this type of speculation could remain in the discussion where it is more relevant.

Ok, in the revised manuscript we exclude the references about the anthropogenic forcing by deleting the first sentence in the Abstract and the first paragraph in the Introduction.

L109: "for the first time to our knowledge in this region" - this is a broad claim and not necessary. Ok. Deleted.

Section 3.1: Please add significant detail on the mixed layer estimation method.

Right, it has been added in the revised manuscript as indicated at the beginning of this document.

L222: The 2018 profiles with deep mixed layers are not in the Irminger Sea.

Right, as explained above, we changed the limit of the pink box and refer to our pink box as SECF.

L237: "Water masses formed are very similar" It should at least be acknowledged that they are formed much closer to the Labrador Sea than in previous years.

Right, when excluding the floats south of Cape Farewell as requested by referee 3, the properties of the water mass formed in the SECF region in W2018 is not similar to the formed in the Labrador Sea in W2018. The sentence "Water masses formed are very similar" is excluded in the new version of the manuscript.

L247: maybe instead: "heat alone" at the end of this sentence. Ok

L248: This paragraph is confusing. Perhaps referring to Figure 4 earlier on would help?

Ok, thank you for noting it. The objective of the paragraph was to show that SFek cannot be neglected in BFek. We present this point more clearly in the revision. Morevoer, in this section, we add a paragraph describing Figure 4.

L331: "despite they were also fresher" ! "despite the fact that they were also fresher"

Ok, we will change it.

L340: Refer to figure 6. Ok, we will write, "The predicted convection depths are determined as the depth at which B(zi) (Fig. 6a), equals the atmospheric forcing."

L348: Clarify what happened here. These floats only profiled down to 1,100m?

Exactly. We would rewrite the sentence as: "This result is in line with the fact that among the 10 profiles that we used to compute Q3 in W2018, 6 showed deep convection down to 1,100 m and were recorded by floats **with a maximum profiling depth** of 1,100 m, most likely leading to an underestimation of the MLD." However, this sentence is going to be deleted in the revised manuscript. In the revised paper, and following the suggestion of referee 3, we exclude from the analysis the profiles that do not extend beneath the base of the mixed layer, because it results in bias in the properties related to the mixed layer.

L351: I was also confused by the fact that the author's claim to neglect advection, but cite advection of properties from the Labrador Sea as a reason for favorable preconditioning. Perhaps remove that claim. Additionally, the fact that the T and S properties are not homogeneous goes against the idea that deep convection is occurring locally.
We agree that this paragraph was confusing. We now identify lateral advection as a possible cause for the buoyancy budget residuals. The profiles with non-homogenous TS in the mixed layers are now excluded from the analysis.

L370: hydrological ! hydrographic. Yes, hydrographic, we will change it.

L370: anomalies relative to what? Related to the mean 2002 – 2016, we added it in the text.

L383: Why would only the properties in the 1200-1400 depth range be advected? Or are they the only ones that have a profound effect? See above. Please clarify.
See answer in your comment above.

L415/Figure 10: Not sure how this figure and paragraph are linked to the rest of the study.
This figure and paragraph were not essential for the conclusions of the paper. We decide to remove them in the revised manuscript.

L470: hydrological ! hydrographic Yes, hydrographic, we will change it.

Figure 4: Note the differences between the axis ranges in the caption. This figure could be featured earlier as it provides important context.
Ok, we will write in the figure caption: "Note the differences between the axis ranges". We refer to this figure earlier in the section 4.2 of the revised manuscript.

Figure 5: From Figure 5d, it appears that the thick density layers are actually below the densities that are being ventilated in the Irminger Sea (white areas). This supports the idea that they are being advected from the Labrador Sea.
In winter 2015 and 2016 the thick density layers have a density of 32.37 Kg m$^3$, that corresponds to sig0 equals to 27.746 Kg m$^3$ which is the density of the mixed layer in the SECF (Fig. 3 in the manuscript). In winter 2017 and 2018 the thick density layers are found at denser density (32.38 Kg m$^3$), that corresponds to sig0 equals to 27.754 Kg m$^3$ which is the density of the mixed layer in the SECF for these winters (Fig. 3). These results support local formation.
Accordingly, we add at the end of the first paragraph of section 4.3: "The denser density of the core of the thick layers in 2017 -2018 compared with 2015 - 2016 agrees with the densification of the mixed layer SECF shown in Table 1 and Fig. 3."

Figure 6: Please clarify: are you using all Argo data within the box, or only the ones with deep mixed layers?
All data. To clarify, the Figure caption will be modified as, "they were calculated from **all** Argo data measured in the Irminger box (see Fig. 1) in September before the winter indicated in the legend."

Figure 7: This is a very interesting figure! Could feature more prominently and be usedto describe some key differences between the Labrador and Irminger Seas.
Right, we used this figure in the discussion (lines 394 -414) when comparing the preconditioning in the Labrador Sea and in SECF.

Reddish! red. Bluish ! blue.Ok, in the revised manuscript we change the figure caption of this figure.

Figure 8: missing a) b) c) labels on the figure. Ok, we add them.

---

## Author Comment (AC3) · 7 Oct 2019

Two superimposed cold and fresh anomalies enhanced Irminger Sea deep convection in 2016 - 2018
by Patricia Zunino, Herlé Mercier, and Virginie Thierry

In this manuscript persistence of deep convection in the Irminger Sea is investigated. One winter of particularly severe atmospheric forcing and deep convection was followed by three winters of climatological strength which also had deep mixed layers. The authors quantified the buoyancy loss required for deep convection to commence each winter and concluded that the preconditioning arising from the previous winter's homogenization of the water column was a main reason for the persistence of deep convection.

I think this manuscript has the potential to be an important and valuable contribution to better understand deep water formation in the Irminger Sea /subpolar North Atlantic. However, as made clear also by the other reviewers, I have concerns about the determination of mixed-layer depths. As such, I recommend that the paper be revised before publication.

Thank you for your valuable comments; they help us improve our work. In the following we answer point by point to all of your comments and explain how we will modify the manuscript accordingly.

**Major comments:**

I am not convinced that automated routines, such as the threshold or split and merge methods, are particularly suitable for determining the vertical extent of the mixed layer. These routines generally perform well when applied to summer and fall profiles, when the upper ocean is stratified and there is a pronounced density difference between the mixed layer and the lower part of the profile. However, they are less accurate during periods of active convection when stratification is eroded. Furthermore, such routines cannot identify mixed layers that are isolated from the surface, either in the form of vertically stacked mixed layers or by early stages of surface restratification. Such isolated mixed layers are prevalent in the Labrador and Irminger Seas during winter (e.g. Pickart *et al.*, 2002). As pointed out by the other referees, if the density profile is considered in isolation, changes in temperature and salinity may be density-compensated such that the water column can appear to be homogenized while in reality it is not. Examples of that can be seen in Figure 2a-c (in particular 4901809 - 35). To avoid erroneous mixed-layer depths, I strongly recommend employing the semi-objective method developed by Pickart *et al.* (2002) instead of relying on automated routines.

In agreement with the three referees, we have revised our method for estimating MLD. Please see the first part of the response to de Jong (Referee 1) in order to see:
1. the specifications of our revised method for estimating MLD,
2. the comparison of MLD estimated with our revised method and estimated with other methods (de Jong et al., 2012; Pickart et al, 2002).
3. The region and profiles considered for the computation of the characteristics of the MLD (max MLD, Q3, density, temperature, salinity) formed Southeast of Cape Farewell.
4. The similarities and differences between our previous and new estimates.

Deep convection evidently took place in winter 2015 as documented by the many deep mixed layers shown in Figure 1. For winters 2016-2018, on the other hand, the vast majority of the Irminger Sea profiles do not have particularly deep mixed layers. If widespread deep convection occurred also during these winters, there should be many more profiles with deep mixed layers. Is it possible that

the mixed-layer depths determined by the automated routines are remnants of deep convection from a previous winter or from the Labrador Sea where mixed layers are generally deeper?

The percentage of profiles with deep MLD depends on the period during when we compute the statistics. Our previous method was misleading because we considered the *entire* winter for computing the statistics and not only the convection period (see also answer on this point to referee 2). We now identify the period during which deep MLDs > 700 m were observed for each winter in the Southeast Cape Farewell (SECF) region (pink box in Fig. R4 in referee 1 answer) (see answer to reviewer 2 for more details). Then, we quantified the percentage of floats that measured deep MLD in the region and during the period of deep convection. The results are shown in table R2. The lower % is found for winter 2017, but it is still substantial and reflects the fact by the fact that the floats showing deep MLD were found southwest of the SECF box suggesting that convection did not occur over the full box. The results of this sensitive study will be added to the section 4.1 of the revised manuscript.

Table R2. Sensitivity study about the Argo float coverage in the SECF region (pink box in Figure R4 in the answer to referee 1). Period is the period during which floats with deep mixed layers were observed. We indicate the total number of floats found in the SECF region during the indicated period, and the number of floats showing deep convection. Finally, the percentage of floats showing deep convection is indicated.

| | Deep convection period | n floats in the region | n floats in the region with deep convection | % of floats in the region with deep convection |
|---|---|---|---|---|
| W2015 | 15/01/2015 to 21/04/2015 | 11 | 8 | 73% |
| W2016 | 22/02/2016 to 21/03/2016 | 4 | 2 | 50% |
| W2017 | 16/03/2017 to 04/04/2017 | 9 | 3 | 33% |
| W2018 | 24/02/2018 to 26/03/2018 | 4 | 2 | 50% |

We do not think that the observed MLD are remnants of deep convection from a previous winter or from the Labrador Sea because the new estimates of MLD are from profiles homogenous in terms of density, temperature and salinity. Most importantly, the fact that the 1D-buoyancy budget is nearly closed (section 4.3) is also an indication that deep convection occurred locally in the SECF box during winters 2016, 2017 and 2018.

To get a more robust estimate of convection in the subpolar North Atlantic these winters, I suggest dispensing with the 700 m "deep convection" criterion and showing if not every mixed layer at least the 50-80% deepest mixed layers encountered by each float every winter. That would remove shallow mixed layers arising from early phases of the seasonal evolution of the mixed layer and profiles obtained within stratified eddies, while the remaining mixed layers would allow for more robust quantification of the general depth of convection.

OK, this seems to be a nice idea, but it would bias low the estimate of convection depth if the statistics of MLD were made using the profiles for the entire winter. The criteria should be applied to the convection period that we select here by considering profiles deeper than 700 m because it is the minimum depth that should be reached for LSW renewal. If apply to those profiles your criteria would not be much different from our Q3. Note that our estimate of convection depth based on the statistical criteria Q3 is equivalent to the aggregate maximal convection depth used by Yashayaev and Loder (2017) and allows direct comparison with this author's results.

Profiles that do not extend beneath the base of the mixed layer (there may be some examples in Figure 2d-f) would result in a shallow bias of the mixed-layer depth estimate and should be excluded from the analysis.
We agree. These profiles located between 48°W and 45°W are not consider in our new results.

Specific comments:
Line 95:
It should be: "...Argo and mooring data..." Corrected

Lines 106 and 361:
Mixed layers exceeding 1400 m depth were determined also from shipboard measurements in the Irminger Sea in April 2015 (Fröb *et al.*, 2016). We add this reference to the revised manuscript.

Line 122:
If the TEOS-10 convention is used, conservative temperature and absolute salinity should be used instead of potential temperature and salinity.
TEOS-10 allows the computation of theta and practical salinity.

Line 123:
Please explain why a salinity of 35 was chosen as a reference value.
This sentence is deleted in the manuscript because we do not use FW in the paper. Sorry for the confusion it may have caused.

Line 124:
Please provide more information about the gridded products. Are different time periods and resolutions the only difference between the products? What are the errors, in particular for the EN4 product which extends back to 1900 and covers some very data-sparse periods?
ISAS and EN4 are optimal interpolation of in situ data, but the optimal interpolation method is not exactly the same in both products due to different choices for the spatial and temporal correlation functions used for the optimal interpolation. Details about both databases are described in the references given in the manuscript (Gaillard et al., 2016; Kolodziejczyk et al., 2017; Good et al. 2013). Note that we used EN4 data from 1993 afterwards and that the monthly temperature and salinity fields at a given time only depends on the data found in a short time window around the date of the analysis. The data sparse-period at the beginning of the 1900 did not influence our results.

Line 130:
Does the net air-sea heat flux include radiative fluxes or only turbulent fluxes?

It includes both radiative and turbulent fluxes. We indicate it in the revised manuscript.

Line 149:
I do not think that 48◦W is commonly used as a border between the Labrador and the Irminger Seas. Many of the deep mixed layers were recorded directly south of Greenland, in a region that is not really part of either the Labrador or the Irminger Seas.
Ok, the limit at 48°W was used just to include in the analysis of the MLD properties the profiles found between 48°W and 45°W in 2018. In the revised computation we used only profiles inside the pink box which limit is at 45°W and we now refer to the pink box as Southeast Cape Farewell (SECF) instead of Irminger Sea. Note that the northern limit of the box is changed from 61°N to 59.3°N. We calculated the atmospheric forcing and the preconditioning considering this new box limit and it does not change the main results and conclusions of our work.

Line 156 and elsewhere:
Please insure that all papers cited in the text are included in the References section. For example is Gill (1982) missing. Ok, thank you for noting it.

Line 174:
How was the depth of the Ekman layer estimated?

We used the Ekman transport and we considered that the SST is representative of the temperature in the Ekman layer. We will clarify this point in the revision.

Line 179:
For consistency, it might be better to use SST also from the EN4 product.
Ok, we have estimated the Ekman Buoyancy Flux (BFek) using EN4 SST.
The horizontal Ekman Buoyancy flux in the SECF region (pink box in Fig. R4 in response to referee 1), accumulated from 1 September to 31 August the year after was estimated with: i) with EN4 SST and EN4 SSS and ii) with ERA SST and EN4 SSS; they are represented in Figure R5. Both time series show the same behavior but the results obtained with EN4 SSS and EN4 SST are smoother than the results obtained with ERA SST and EN4 SSS. Thank you for your comment, we switched to EN4 SST.

[Figure]

Figure R5. Horizontal Ekman Buoyancy flux in the SECF region (56.5° - 59.3°N, 45°W – 38°W), accumulated from 1 September to 31 August the year after estimated: i) with EN4 SSS and EN4SST and ii) with ERA SST and EN4 SSS.

Line 185:

It should be: "...most of the Argo profiles..." Corrected.

Line 197:
It should be: "...to be removed (B(zi)) from the late summer density profile..." Corrected.

Line 234:
Salted, in this context, is not appropriate. "Became saltier" would be a better expression. Corrected.

Line 284:
If B remained nearly constant, does that imply that restratification and advection are unimportant?

It means that the homogeneous layer (600 – 1400 m) formed at the end of winter was not destroyed by the advection by eddies and large scale circulation during the following spring and summer.

Line 297:
Units (m) are missing after 800-1000. Corrected.

Line 321:
What was the basis for choosing the point 59∘N, 40∘W?
Our objective here was to see the evolution of the anomalies in depth and in time. Therefore we choose a point, 59°N, 40°W, in the middle of the box. The result is not sensitive to the location of the point inside the pink box. This information is added to the revised manuscript. In the revised manuscript we present the same figure at 58°N, 40°W, which is centered in the new pink box, instead that at 59°N, 40°W.

Line 377:
If convection exceeded 1400 m in winter 2014-15 (e.g. Fröb *et al.*, 2016), why is it unlikely that this layer was locally formed?
Right, we cannot exclude that the convection of winter 2014-15 cause salinity decrease in the water column. We slightly modified this paragraph in the revised manuscript.

Line 382: The papers by Lavender *et al.* (2000) and Straneo *et al.* (2003) could also be cited here. Ok, we add them to the revised paper.

Line 383: Corroborated is misspelled. Right.

Line 388:
If deep convection occurs every year, perhaps the definition of deep convection should be revised.
This sentence is confusing. In the revised manuscript, this sentence is written as:
"We now compare the atmospheric forcing and the preconditioning of the water column in the SECF region with those of the nearby Labrador Sea where deep convection happens almost every year."

Line 403:

It should be: "...the deep halocline was successively deepening..." Right, thank you.

Line 410:

I am sceptical of the claim that the deepest convection-depth ever observed in the Labrador Sea occurred in winters 2016-2018. Very likely convection in the successive high-NAO winters of the early l990s substantially exceeded convection in winters 2016-2018. At that time mixed-layer depths were at least 2300 m (e.g. Avsic *et al.*, 2006).
Yes, you are right. So, we add to the sentence "since the beginning of the Argo period".

Lines 419 and 421:

Density units are not capitalized consistently. Ok.

Line 425:

It should be: "...observed in both basins..." Right, thank you.

Line 430:

There were no wintertime measurements in the Irminger Sea in the early 1990s, but there is strong indirect evidence that deep convection occurred in the Irminger Sea at that time (see for example publications from the group of R. Pickart).
Right, there are evidences that deep convection occurred in the Irminger Sea in early 1990s (Pickart et al., 2003).
In any case, the three referees find something wrong in this paragraph and Figure 10. Because this paragraph and figure is not important for the conclusions of our paper we decide to remove them in the revised version of the paper.

Line 481:

Acknowledgement is misspelled. Right, thank you.

Lines 519 and 522:

The name de Jong is inconsistently capitalized. Right, corrected.

Figure 5:

Please indicate, for example using tick marks along the top axis, when Argo float profiles were available in the Irminger Sea.
Ok, we add the tick marks in plot 5b.

---

## Author Response (AR2)

Dear Editor,

Thank you for handling our manuscript and providing us with these additional referee's comments. A point by point response to these comments is provided below. The manuscript and the supplementary material with the changes marked-up are also in this document after the referee response.

Best regards,

Patricia Zunino, Herlé Mercier and Virginie Thierry

The authors have tried to address the main comments on their method, previously using a density criterion only to determine a mixed layer and now considering temperature and salinity as well. Their current criterion is still rather wide. The authors state that "The temperature threshold of 0.1°C and the salinity threshold of 0.012 were selected because they correspond to a threshold of 0.01 kg m-3 in density that was previously shown to perform well in the subpolar gyre (Piron et al., 2016)". This is only true if either salinity or temperature is at this threshold, if both are the density difference can be as big as 0.2 kg/m^3.

In her first review of our manuscript, the reviewer noted that our determination of the MLD based only on a density criteria failed when density compensated temperature and salinity anomalies occurred in the mixed layer. The criteria in temperature and salinity that we added in our revision were chosen to address this problem. When temperature and salinity anomalies are density compensated, they do not add together and selecting a criteria for determining the MLD from the temperature or salinity profiles with a threshold equivalent to the one used for density seems correct to us. Visual inspection confirms this. Of course, if temperature and salinity profiles are nearly homogeneous or not compensated in density, the criteria in density sets the MLD, but these were not cases that the reviewer identified as problematic. We added the following sentence in the revised manuscript: "The criteria on temperature and salinity were chosen to perform well when temperature and salinity anomalies within the density-defined mixed layer are density compensated."

The authors attempt to compare their method with earlier publications, however the comparison is made incorrectly stating "Following de Jong et al. method's, three MLD were defined as the depths were the standard deviations were smaller than 0.05 kg m-3 , 0.05°C and 0.005 for density, temperature and salinity, respectively", the error being that the standard deviations were not used as a threshold to determine the bottom of the MLD but as a final check leading to discarding misidentified MLDs (as described in the paper). Since this comparison now features in the supplementary materials it will need to be corrected.

The present manuscript mostly compares the identified MLDs to those we identified in the winter of 2014-2015 (de Jong et al., 2018). For those MLD we used a criteria of "0.015°C for temperature, 0.005 for salinity, and 0.0025 kg m–3 for potential density". "The resulting MLDs of all three variables had to be within 50 dbar to be accepted, after which the temperature-derived MLDs were chosen as final MLDs." Running this criteria on the specific float profiles shown in Figure 2 of the manuscript results in the deepest MLD (of 570 dbar) observed in the profile of float 5904772, cycle 33. For the other profile shown in the Figure 2, and the supplementary material, our methods gives a MLD of only 150 dbar. We made the criteria this strict to sort out actively mixed profiles at the moorings for recently mixed profiles which may have been formed elsewhere and advected to the moorings. Since the authors are looking more of a region than at one specific spot they could make an argument that they can use a wider criterion. In that case I ask that that at least the SI is corrected to show our methods accurately and the difference with earlier estimates be discussed more clearly in the discussion.

We thank the reviewer for noting that. Because of the very strict criteria she used, her method was only adapted to the diagnostic of active mixed layers and thus was not suited to the diagnostic of the MLD our case. Indeed, Straneo et al. (2002) noted that in the ocean interior where baroclinicity is weak there are dynamical reasons for tracer profiles in the mixed layers to be slightly slanted. This confirms that de Jong et al., (2018) criteria, although well suited for their application, are too restrictive in a more general case. Thus we deleted in the supplementary material the comparison between our estimates of MLD and those obtained using de Jong et al.'s method, which was misleading. We noted this information in the supplementary material and we kept the comparison between our results and those obtained using Pickart et al. (2002) method.

In the second paragraph of the results section there is a discussion on the percentage of floats that had deep MLD. This is not very informative, as 1 float with 1 deep ML in the winter of 2017 would give the same percentage as 1 float recording 20 deep MLs in 2015. The percentage of profiles with deep MLs out of all recorded profiles would give a much better indication of the intensity of convection. Overall, this difference in intensity between the winters is mostly neglected in the text ,focusing only on depth (for example lines 364-366 and in the conclusions line 461-467). This is a missed opportunity since the authors have already investigated the likely cause for this difference in intensity, being weaker surface fluxes in the winters after 2014-2015.

Thank you for noting that. We now emphasize this result in the conclusion: "The deep convection of W2015 was observed over a larger region and during a longer period of time than the deep convection events of winters 2016, 2017 and 2018." Otherwise, the percentage of profiles with deep MLs out of all recorded profiles would not necessarily give a better indication of deep convection intensity because spatial and time samplings are not homogeneous in the studied region. We believe that bias due to sampling is less with statistics based on the number of floats.

Line by line comments
Line 50: the "exceptional" here is disputable because we don't have a long enough record to say whether it is really that unusual and because there is such a distinct difference in intensity over the four winters.
Ok, we removed "exceptional" in line 50 of the abstract.

Line 156: The use of Q and Q3 as acronyms for parameters that are not associated is confusing, especially if they occur in subsequent sentences. Consider using a different acronym for Q3. How useful is the third quartile if there are only three profiles with deep MLDs in a winter?

We used Q3 because it is the usual abbreviation for the third quantile in statistic. As explained at the end of section 3.1, Q3 is equivalent to the aggregate maximum depth of convection defined by Yashayaev and Loder (2016). In the revised version of the manuscript we changed the "Q3" to "the aggregate maximum depth of convection".

We consider that Q3 estimate is appropriate because it depends on the values of all the values of the population and is not just the maximum value. Moreover, it allows direct comparison with the results of Yashayaev and Loder (2016) and Piron et al. (2017) who used this estimator.

Line 199: The Monte Carlo method does not address the bias in a reanalysis product as a whole, as observed by Josey et al.

This is correct.

Line 259: "Despite Bsurf* is mainly explained by Q, the accumulated FWF* amounts to ~10 % of the accumulated Q with opposite sign. The air-sea buoyancy flux is 10% lower on average than the air-sea heat flux." This seems to be saying the same thing twice. Consider rephrasing.

Thank you for noting it, we rephrased it as: "$B_{surf}$* is 10 % lower on average than Q because of the buoyancy addition by FWF*."

Lines 330-340: The discussion of anomalies here would be more interesting/easier to follow if it was immediately linked to local convection and advection pattern as described in lines 393-397.

We are sorry, but we disagree. We believe that lines 393-397 fit well in the discussion section.

**References**

Straneo, F., M. Kawase, and R. Pickart: Effects of Wind on Convection in Strongly and Weakly Baroclinic Flows with Application to the Labrador Sea, Journal of Physical Oceanography, V32, *https://doi.org/10.1175/1520-0485-32.9.2603*, 2002.

[revised manuscript text omitted]

SUPPLEMENTARY MATERIAL
S1. METHODS FOR ESTIMATING THE MIXED LAYER DEPTH

In this paper, mixed layer depth (MLD) was estimated using the threshold method described in
section 3.1. Our estimates were compared to those based on the methods of
Pickart et al. (2002), which like ours is adapted to slightly slanted tracer profiles in the mixed
layers as those often observed in the central subpolar gyre (Straneo et al. 2002). Pickart et al. (2002)
requires a first guess for the mixed layer

Pickart et al. (2002)

that we have taken equal to the MLD estimate obtained with our threshold method (section
3.1 of this paper) . Then, the mean and standard deviation of the $\sigma$, S and
$\theta$ were estimated from the surface to the initially defined MLD. Finally, the two–standard deviation
envelopes overlaid on the original profile were plotted on the $\sigma$, S and $\theta$ profiles. The mixed layer
depth was determined as the location where the profile permanently crossed outside of the two–
standard deviation envelope.

S.2. FIGURES IN SUPPLEMENTARY MATERIAL

[Figure]

**Figure S1**. Comparison of MLD estimated for float 6901171 – 101 by our method (black point) and by Pickart et al.'s method (horizontal discontinuous gray line). The continuous colored lines are the vertical profiles of σ, S and θ measured by the Argo float. The dashed colored lines are the two–standard deviation envelope considered in the Pickart et al.'s method.

[Figure]

**Figure S2**. The same as Fig. S1 but for profiles 59004772 – 33.

[Figure]

**Figure S3.** Mean (1993 - 2016) seasonal cycle of air-sea flux of buoyancy (Bsurf*), heat (Q) and freshwater (FWF*) averaged on the SECF region (pink box in Fig. 1). Data origin: ERA-Interim, accumulated every 24h.

[Figure]

**Figure S4**. Time series of accumulated (from 1 September to 31 March the year after) buoyancy air-sea flux ($B_{surf}$) and buoyancy Ekman flux ($BF_{ek}$) and the sum of both. The year in the x-axes indicates the flux accumulated from 1 September y-1 to March y.

[Figure]

**Figure S5**. Number of Argo profiles by year used in Figure 5.

[Figure]

**Figure S6**. Annual anomalies of salinity in the surface layer (20 – 100 m) estimated from ISAS

database. Reference period: 2002 – 2016. We represented only anomalies larger than one standard deviation of the mean.